# DISENTANGLING LATENT EMBEDDINGS WITH SPARSE LINEAR CONCEPT SUBSPACES (SLiCS)

## ABSTRACT

Vision-language co-embedding networks, such as CLIP, provide a latent embedding space with semantic information that is useful for downstream tasks. We hypothesize that the embedding space can be disentangled to separate the information on the content of complex scenes by decomposing the embedding into multiple concept-specific component vectors that lie in different subspaces. We propose a supervised dictionary learning approach to estimate a linear synthesis model consisting of sparse, non-negative combinations of groups of vectors in the dictionary (atoms), whose group-wise activity matches the multi-label information. Each concept-specific component is a non-negative combination of atoms associated to a label. The group-structured dictionary is optimized through a novel alternating optimization with guaranteed convergence. Exploiting the text co-embeddings, we detail how semantically meaningful descriptions can be found based on text embeddings of words best approximated by a concept's group of atoms, and unsupervised dictionary learning can exploit zero-shot classification of training set images using the text embeddings of concept labels to provide instance-wise multi-labels. We apply SLiCS to CLIP embeddings, the highly-compressed autoencoder embeddings from TiTok, and the latent embedding from self-supervised DINOv2. We show that the disentangled embeddings provided by our sparse linear concept subspaces (SLiCS) enable concept-filtered image retrieval that is more precise. In addition to conditional generation using image-to-prompt from the components, we develop a model for conditional generation from each concept subspace using diffusion posterior sampling. Quantitative and qualitative results highlight the improved precision of the concept-filtered image retrieval for all embeddings.

## 1 INTRODUCTION

Deep vision-language co-embedding models trained on large datasets of captioned images capture both visual and semantic features from the original inputs, encoding them into a dense vector space to generate image and text embeddings. These models are widely applied in tasks such as cross-modal retrieval and conditional generative modeling. In particular, the pre-trained CLIP model (Contrastive Language-Image Pre-Training) (Radford et al., 2021) uses separate image and text encoders to embed inputs into the same vector space using contrastive learning (Poole et al., 2019; Gutmann & Hyvärinen, 2010). Deep vision embedding models trained using self-supervised objectives (Chen et al., 2020b;a; 2021; Caron et al., 2020; Oquab et al., 2023) and high-quality autoencoders (Van Den Oord et al., 2017; Esser et al., 2021) with more compressed embeddings (Yu et al., 2024) also provide semantically rich embeddings (Beyer et al., 2025).

Even though the embeddings are semantically rich, the dimensions of the dense vectors (or tokens) are not directly interpretable. For complex scene imagery, the ability to decompose an embedding vector into components corresponding to interpretable concepts would enable *concept-filtered* retrieval and conditional generation. Towards this goal, we propose Sparse Linear Concept Subspaces (SLiCS) to disentangle a given embedding vector into a set of component vectors, each corresponding to a concept identified as active. Each component is a non-negative combination of basis vectors (atoms) defining a subspace for a coherent concept. The disentangled components provide an understanding of the distinct concepts present in an image. That is, each component subspace (actually

a cone) captures the possible variation of its associated concept within the embedding space. In the case of vision-language embeddings like CLIP, we exploit the fact that the image and text embedding spaces are aligned and select descriptive words for each concept by using the word embeddings that are well-reconstructed by the concept's atoms. We also propose a novel use of diffusion posterior sampling (DPS) (Chung et al., 2022) to sample from the concept subspaces.

We assume that a deep image embedding encompasses the entire scene, densely preserving information about every object and background element. In retrieval, this holistic representation of the query image may not be ideal when we are interested in matching only a subset of aspects of the query image, as irrelevant aspects may adversely affect the retrieval. Suppose a query image contains multiple distinct concepts; then, assuming compositionality, the embedding can be decomposed so that each concept-labeled component can be used to retrieve images similar to the aspects of the query image that are contained within that concept. For example, consider a complex query image containing a dog; to retrieve other images containing similar dogs, the "animal" component of the image embedding can be used to query the image database. Without filtering, another image that has a mostly similar scene but is missing the "dog" may be very similar in the latent space. Thus, concept-filtered retrieval, where the user can define which concepts should be focused on, has the potential to make image retrieval more flexible and precise.

Given a concept-labeled image dataset, SLiCS learns a subspace for each concept as a supervised dictionary learning problem, wherein multiple concepts may be co-active and compete to explain the embedding. For vision-language co-embedding models, the labels can be obtained from zero-shot classification of text embeddings of concepts. This provides for an unsupervised version of SLiCS. However, SLiCS does not rely on the presence of text embeddings for supervised training, and any vision embedding can benefit from SLiCS-based disentangling for more precise retrieval. In summary, we propose SLiCS as an approach for learning to disentangle dense embeddings to enable concept-filtered retrieval and image generation in cases where the embedding can be directly decoded or used as a conditioning prompt. Experimental results show that SLiCS provides better retrieval performance compared to unfiltered retrieval across three different embedding models. An analysis of the geometry of the concept subspaces and the distribution of components shows that there is a balance between capturing the variance of each concept and disentangling distinct subspaces. Ablation studies show that non-negativity and learning with co-active concepts improves the precision of the retrieval. Finally, the correlation between coefficients associated to the atoms of concepts reveals that the usage of atoms within a concept are often correlated.

## 2 RELATED WORK

CLIP is leveraged in a number of image retrieval methods using text queries (Chaudhary et al., 2020) or multi-modal approaches (Baldrati et al., 2022; Couairon et al., 2022; Liu et al., 2021) that use a combination of image and text to retrieve images similar to the contents of a query image modified by the provided query text (Wei et al., 2024; Jiang et al., 2025). However, directly decomposing image embeddings to perform concept-filtered retrieval remains unexplored. Gandelsman et al. (2024) explore the interpretability of the intermediate representations in CLIP, breaking the vision transformer image encoder into patches and attention heads, trying to find the text description for each attention head. However, it does not directly handle decomposition.

SpLiCE (Bhalla et al., 2024) decomposes an image embedding on a set of fixed word embeddings with sparse coding. It operates in an unsupervised fashion, but the lack of group structure limits the interpretation to instance level. More similar to our approach, Kobs et al. (2023) use PCA to create a subspace with the guidance of a set of word embeddings that describe the same concept or aspect. Compared with these methods, SLiCS operates solely on image embeddings, only using text embeddings to construct pseudo-labels or enable interpretation of the decomposition.

Underlying SLiCS is a supervised form of constrained dictionary learning, combining non-negativity of coefficients (Ding et al., 2008) with an assumption of non-overlapping group structure (Szabó et al., 2011; Zelnik-Manor et al., 2012). The group structure makes SLiCS distinct from reconstruction-based disentanglement of latent spaces, such as $\beta$-VAE (Higgins et al., 2017) or sparse autoencoders (Huben et al., 2024), which require post-hoc interpretation.

SLiCS is related, but distinct, from other methods for interpreting latent embedding using known (or inferred) concepts. Notably, other work's concepts are often *attributes* of one or more classes, for example 'has stripes', whereas we focus on concepts that are often the classes of foreground objects or a classification label of the scene itself ('outdoor' and 'indoor'). In particular, SLiCS can be contrasted with Concept Bottleneck Models (CBMs) (Koh et al., 2020) and concept vectors (Kim et al., 2018) by the modeling principle—SLiCS is a generative model of the latent embedding, whereas CBMs and concept vectors are discriminative. CBMs are trained to predict concept occurrence, and their predictions are used for downstream tasks. Concept vectors are simply the normal vector of hyperplanes of linear concept classifiers using the activation of a neural network layer that was directly trained for the downstream task. Interpretable Basis Decomposition (Zhou et al., 2018) first learns concept vectors from the penultimate layer, then decomposes the final layer weight for each class as a non-negative linear combination of these vectors. Concept whitening (Chen et al., 2020c) is an approach to replace a layer of a neural network to encourage the whitened activation vectors across the neurons to align with a unit vector for the concept, when the concept is active, ensuring the network activations more interpretable. While useful for interpretation, none of these methods decompose embeddings to capture the variation within a concept. Therefore, they cannot enable concept-filtered retrieval using the original embedding representation of the images in the pool, nor enable conditional generation.

The methods most similar to SLiCS are Concept Activation Vector (CAV) models that use non-negative matrix factorization (NMF) to learn patterns of the non-negative activation vectors for a given class (Zhang et al., 2021; Fel et al., 2023). SLiCS differs in that it does not require non-negative embeddings, and it explicitly models co-activation of different concepts, enabling disentanglement (Erogullari et al., 2025). Our supervised SVD (S-SVD) baseline, which does not model the co-activation of concepts, provides a baseline without disentanglement.

Beyond labeled concepts, prior work has shown how concepts could be automatically extracted from clustering of image segments (Ghorbani et al., 2019), or learned for completeness with inductive priors (Yeh et al., 2020). With the advent of co-embedding models like CLIP, Yuksekgonul et al. (2023) showed how zero-shot labels of relevant concepts for a task sourced from ConceptNet (Speer et al., 2017) could be used to learn concept vectors.

## 3 METHOD

Our proposed method, SLiCS, learns to disentangle vectors by decomposing them into a sparse sum of components lying in "subspaces" that are in fact positive cones, each defined by a non-negative combination of a group of vectors organized into a dictionary. When all the coefficients of a component group are zero, the component is zero, and the concept is inactive. The group structure enforced on the vectors makes the decomposition more structurally interpretable at the concept level, while at the same time maintaining the diversity within each disentangled subspace. In the supervised case with multi-label classes, the coefficients are forced to be inactive if a class is not present. Even with this known group activity, both the coefficients and the vector groups need to be learned. We propose a novel supervised dictionary learning algorithm with non-negative coefficients, inspired by the K-SVD algorithm (Aharon et al., 2006), which exploits rank-1 approximations to simultaneously update an atom and its contributions.

### 3.1 SPARSE LINEAR CONCEPT SUBSPACES: LINEAR SYNTHESIS MODEL

Let $\phi : \mathcal{X} \to \mathbb{R}^d$ denote a pre-trained neural network that maps a query image $X_\star \in \mathcal{X}$ to an embedding $\mathbf{x}_\star = \phi(X_\star) \in \mathbb{R}^d$. For $S$ concepts, the embedding $\mathbf{x}_\star$ is approximated as

$$\mathbf{x}_\star \approx \sum_{j=1}^{S} \mathbf{v}_j^\star = \sum_{j=1}^{S} \sum_{i=1}^{M_j} \alpha_{j,i} \mathbf{b}_{j,i} = \sum_{j=1}^{S} \mathbf{B}_j \boldsymbol{\alpha}_j = \mathbf{B}\boldsymbol{\alpha}, \tag{1}$$

where $\mathbf{v}_j^\star \in \mathcal{V}_j \subseteq \mathbb{R}^d$ is the component residing in the corresponding positive cone $\mathcal{V}_j$, $\mathbf{B}_j = [\mathbf{b}_{j,1}, \ldots, \mathbf{b}_{j,M_j}] \in \mathbb{R}^{d \times M_j}$ is the dictionary, each $\mathbf{b}_{j,i}, \quad i \in \{1, \ldots, M_j\}$ is an atom, and $\boldsymbol{\alpha}_j \in \mathbb{R}_{\geq 0}^{M_j}$ is the non-negative coefficient vector, all associated to the $j$th concept. The combined dictionary is $\mathbf{B} = [\mathbf{B}_j]_{j=1}^{S} \in \mathbb{R}^{d \times M}$, with $M = \sum_{j=1}^{S} M_j$. The combined coefficient vector

$\boldsymbol{\alpha} = [\boldsymbol{\alpha}_j]_{j=1}^S \in \mathbb{R}_{\geq 0}^M$ has a group sparse pattern: $\boldsymbol{\alpha}_j = \mathbf{0}$ implies that $\mathbf{v}_j^\star = \mathbf{0}$ and the $j$th concept is inactive, whereas if the $j$th concept is active $\|\mathbf{v}_j^\star\| > 0$ and $\|\boldsymbol{\alpha}_j\| > 0$. Hence, the support of $\boldsymbol{\alpha}$ indicates the active subspaces in $\mathbf{x}_\star$.

Here $\boldsymbol{\alpha}_j$ is constrained to be non-negative, which is motivated by the fact that cosine similarity is used by CLIP to measure the similarity of embedding vectors. Embedding vectors pointing in opposite directions may represent semantic negation or dissimilar input images. Thus, an image embedding $\mathbf{x}_\star$ is considered to semantically include a concept only if its corresponding coefficient is positive. Furthermore, the enforcement of non-negativity reduces the "subspaces" to positive cones.[1] For embeddings normalized to lie on a hypersphere, one can visualize the intersection of the positive cone and the hypersphere as a spherical polygon, as illustrated in a three-dimensional latent space case in Fig. 1 for $S = 3$. If $\mathbf{x}_\star$ lies in one of these spherical polygons, that means that it could

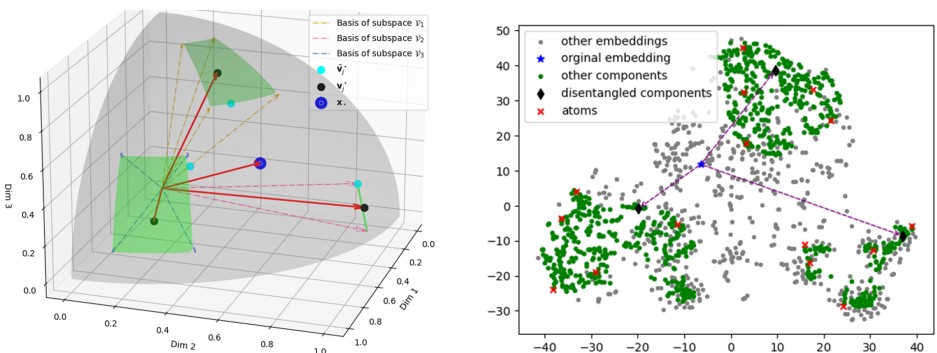

Figure 1: (Left) Illustration of the "subspace" disentanglement of a three-dimensional latent space when $S = 3$. The gray surface denotes the unit sphere. The "subspaces" are the green surfaces, where the first and third concepts are defined by $M_1 = M_3 = 4$ atoms and the second is defined by $M_2 = 2$ atoms. The blue dot denotes $\mathbf{x}_\star$. The three black dots denote $\mathbf{v}_1^\star, \mathbf{v}_2^\star, \mathbf{v}_3^\star$. The three cyan dots denote $\tilde{\mathbf{v}}_1^\star, \tilde{\mathbf{v}}_2^\star, \tilde{\mathbf{v}}_3^\star$, which are the nearest points to $\mathbf{x}_\star$ in the concept subspace. (Right) T-SNE visualization of disentangled subspaces by supervised SLiCS. An embedding of one point and its three components, lying in the three subspaces each defined by 5 atoms, are marked.

be described by a single active concept such that the atoms of the concept can fully describe it. An embedding vector of an image with multiple concepts could lie outside of these spherical polygons and be approximated by a non-negative combination of component vectors from multiple spherical polygons. To illustrate this for a high-dimensional embedding space a t-SNE visualization (Van der Maaten & Hinton, 2008) of actual embeddings, their components, and atoms is shown in Fig. 1.

### 3.2 SPARSE LINEAR CONCEPT SUBSPACES: DISENTANGLEMENT

Given the group-structured dictionary and the set of active concepts $\mathcal{J}$, the corresponding set of components can be obtained from $\mathbf{x}_\star$ by solving a non-negative least squares (NNLS) to obtain the coefficients.

$$\mathbf{v}_j^\star = \mathbf{B}_j \boldsymbol{\alpha}_j, \text{ where } [\boldsymbol{\alpha}_j]_{j=1}^S = \underset{[\mathbf{a}_j]_{j=1}^S \in \mathbb{R}_{\geq 0}^M}{\arg\min} \|\mathbf{x}_\star - \sum_{j \in \mathcal{J}} \mathbf{B}_j \mathbf{a}_j\|_2, \quad \text{s.t. } \|\mathbf{a}_j\| = 0, \quad j \notin \mathcal{J}. \tag{2}$$

NNLS is a convex quadratic program with linear constraints enforcing the non-negativity, which also yield sparsity among the coefficients (Slawski & Hein, 2013). Standard algorithms (Lawson & Hanson, 1995) typically scale quadratically with the number of atoms.

When a specific concept's component is desired, as in concept-filtered retrieval, one can perform a partial disentanglement by approximating $\mathbf{x}_\star$ using only $\mathbf{B}_j$. This projects $\mathbf{x}_\star$ to the $j$th concept's "subspace", yielding the vector in the subspace closest to the $\mathbf{x}_\star$,

$$\tilde{\mathbf{v}}_j^\star = \mathbf{B}_j \boldsymbol{\alpha}_j = \underset{\mathbf{v} \in \mathcal{V}_j}{\arg\min} \|\mathbf{x}_\star - \mathbf{v}\|_2, \text{ where } \boldsymbol{\alpha}_j = \underset{\mathbf{a} \geq \mathbf{0}}{\arg\min} \|\mathbf{x}_\star - \mathbf{B}_j \mathbf{a}\|_2. \tag{3}$$

---

[1]Out of convenience, we keep referring to the positive cones as "subspaces" in the following part of the work.

If different concept dictionaries are not orthogonal, $\tilde{\mathbf{v}}_j^\star$ is generally different than $\mathbf{v}_j^\star$, as the latter considers all active concepts. This is illustrated in Fig. 1(a). In addition to being simpler, partial disentanglement performs better for concept-filtered retrieval, since $\tilde{\mathbf{v}}_j^\star$ lies closer to $x_\star$ and preserves more information in it (as shown in Fig. 6 in the Appendix).

When the active concepts are unknown, but only a few are assumed to be active, one can apply sparse coding algorithms with additional non-negativity constraints to estimate the unknown coefficients. In particular, Orthogonal Matching Pursuit (Mallat & Zhang, 1993; Rubinstein et al., 2008) or a group-structured variant (Swirszcz et al., 2009) are greedy approaches that iteratively add individual atoms or concepts, respectively, updating the coefficients after inclusion.

### 3.3 Supervised Dictionary Learning with Non-negative Coefficients

We now consider a supervised dictionary learning for SLiCS using a training set with labels $\{(X_i, \mathbf{y}_i)\}_{i=1}^N$, where $\mathbf{y}_i \in \{0,1\}^S$ is a $S$-dimensional binary label vector that indicates the presence of each concept in image $X_i \in \mathcal{X}$. $\mathbf{X} = [\mathbf{x}_i]_{i=1}^N \in \mathbb{R}^{d \times N}$ is the concatenation of the training set mapped to the embedding space, $\mathbf{x}_i = \phi(X_i)$. Let $\mathcal{I}_j = \{i : y_{ij} = 1\}$ denote the index set that contains the indices for which concept $j$ is present. Supervised SLiCS minimizes the mean squared reconstruction error of the approximation equation 1, using $\mathbf{A} = [\boldsymbol{\alpha}^i]_{i=1}^N \in \mathbb{R}_{\geq 0}^{M \times N}$ to denote the coefficient matrix, the problem can be compactly written as

$$\min_{\mathbf{B}, \mathbf{A} \geq 0} \quad \|\mathbf{X} - \mathbf{BA}\|_F^2 / N \quad \text{s.t.} \quad \boldsymbol{\alpha}_j^i = \mathbf{0} \quad \text{if} \quad y_{ij} = 0. \tag{4}$$

To solve this problem, we develop a supervised dictionary learning algorithm under a non-negative coefficient constraint, based on the K-SVD algorithm (Aharon et al., 2006).[2]

In our approach, we enforce group sparsity according to the labels and enforce non-negativity constraints on $\boldsymbol{\alpha}^i$. The solution is given by non-negative least squares (NNLS) on the supported concepts

$$\boldsymbol{\alpha}^i = \operatorname*{arg\,min}_{[\mathbf{a}_j]_{j=1}^S \in \mathbb{R}_{\geq 0}^M} \|\mathbf{x}_i - \sum_{j=1}^S \mathbf{B}_j \mathbf{a}_j\|_2, \quad \text{s.t.} \quad (1 - y_{ij})\|\mathbf{a}_j\| = 0, \quad j \in \{1, \ldots, S\}, \tag{5}$$

where the constraint ensures that the coefficients are zero for inactive concepts $y_{ij} = 0$. Given the coefficients $\mathbf{A} = [\boldsymbol{\alpha}^i]_{i=1}^N$, the optimization problem for the $m$th atom is

$$\min_{\mathbf{b}} \sum_{l \in \mathcal{L}_m} \min_{\beta_l \geq 0} \|\mathbf{x}_l - \sum_{k \neq m} \mathbf{b}_k A_{kl} - \beta_l \mathbf{b}\|_2^2 = \min_{\mathbf{b}, \boldsymbol{\beta} \geq \mathbf{0}} \|\mathbf{E} - \mathbf{b}\boldsymbol{\beta}\|_F^2, \tag{6}$$

where $\mathbf{e}_l = \mathbf{x}_l - \sum_{k \neq m} \mathbf{b}_k A_{kl}$, $\mathbf{E} = [\mathbf{e}_l]_{l \in \mathcal{L}_m}$ is the residual error matrix, and $\mathcal{L}_m = \{l \in \{1, \ldots, N\} : A_{ml} \neq 0\}$ is the subset of the training set where the atom is active. Without the non-negativity constraints, the rank-1 truncated SVD of the residual error provides the optimal update of the atom and coefficients, by the Eckart–Young–Mirsky theorem. However, as this does not ensure non-negativity of coefficients it may no longer be the optimal solution to minimize the reconstruction error. In fact, finding an optimal solution with non-negativity—a rank-1 semi-non-negative matrix factorization problem—is NP-Hard (Gillis & Kumar, 2015). Nonetheless, we start from the rank-1 SVD, selecting the polarity of the atom through a simple majority rule that we prove is guaranteed to be the optimal polarity (details are given in Appendix A.1.1), and apply thresholding to ensure coefficients are non-negative. While this SVD-based update is still heuristic, the simultaneous update of atom and coefficient converges, as we show in Appendix A.1.2. Empirically, the simultaneous update is more efficient than an alternating update we discuss in Appendix A.1.3.

For initialization, we apply truncated SVD to embeddings of the $j$th concept $[\mathbf{x}_i]_{i \in \mathcal{I}_j} = \mathbf{U}^L \boldsymbol{\Sigma} \mathbf{U}^{R\top}$, taking the columns of $\mathbf{U}^L$ corresponding to the top $M_j$ singular values, choosing the optimal polarity for each, to initialize the $j$th block $\mathbf{B}_j$.[3] One could select $M_j$ to ensure a maximum error fraction

---

[2]K-SVD is an unsupervised dictionary learning algorithm that iteratively solves $\mathbf{A}$ using greedy sparse coding algorithms, such as orthogonal matching pursuit (Pati et al., 1993), and solves $\mathbf{B}$ by separately updating each atom using a rank-1 truncated SVD of the residual error matrix formed from the subset of the training set where the atom is active by removing the atom's contribution to the approximation, while fixing the contribution of other atoms. This update also simultaneously updates the corresponding coefficients of the updated atom.

[3]One could also initialize each block in the dictionary in the same way as K-means is initialized, using random samples from $\mathcal{I}_j$ as the atoms in the $j$th block. We test two different methods, and in practice, the SVD-based approach performs consistently better in terms of the approximation error of the final dictionary.

(equivalently, a minimum proportion of variance explained) as is done in PCA. Another option would be to set $M_j$ proportional to $|\mathcal{I}_j|$. For simplicity, we assume all dictionaries have the same size $d_0 = M_j, \quad j \in \{1, \ldots, S\}$ ($M = S \cdot d_0$ total atoms) and vary $d_0$. Finally, we run for a fixed number $T$ of updates, but a stopping criterion based on error reduction (possibly on a validation set) or the norm of the difference of atom updates can be used. The complete set of steps is described in Alg. 1 in Appendix A.1.

If the training set of embeddings is too large to fit in memory, it can be broken into disjoint mini-batches. If $\mathcal{B} \subset \{1, \ldots, N\}$ is a batch of indices, then coefficients are estimated on this batch $[\boldsymbol{\alpha}^i]_{i \in \mathcal{B}}$ and then all atoms (and corresponding coefficients) that are active on the batch are updated. The training set is shuffled among the batches after each epoch.

### 3.4 Unsupervised Dictionary Learning via Zero-shot Concept Classification

While the labels provide direct guidance on the group sparsity in the supervised cases, we also implement an essentially unsupervised version by exploiting the image-text alignment of CLIP if the set of concepts and the expected number of active concepts are known. Suppose we have the CLIP text embeddings $\{\mathbf{w}_i\}_i^S$ of the $S$ concept words across the templates (for example, "This is a picture of {word}"). The cosine similarity between an image embedding and each text embedding serves as a measurement of how close the word is to the image. In zero-shot classification, the concept with the highest score is the prediction. Here, since there are multiple concepts presented in each image we pick the highest $\tilde{S}$ concepts as active. Alternatively, a variable number of active concepts per instance could be selected. Given the pseudo-labels, unsupervised SLiCS uses the same algorithm as the supervised one.

### 3.5 Disentanglement for Concept-filtered Retrieval

One of the advantageous applications of SLiCS is the ability to conduct concept-filtered image retrieval. Accessing latent embeddings to measure the similarity between two images is easy and computationally cheap. When using a multilabel image as the query, by applying SLiCS to disentangle the embedding of the query, one can exploit a concept-filtered component to retrieve images based on a similarity scope focusing on a specific concept. Given a query embedding $\mathbf{x}_\star$, concept-filtered retrieval based on the $j$th component uses the similarity score $r_\star^j = \cos(\tilde{\mathbf{v}}_j^\star, \mathbf{x})$ for ranking a candidate image $\mathbf{x}$. The retrieval is based on ranking by sorting the pool of candidate images in descending order based on the similarity scores. As a baseline, unfiltered retrieval (e.g., UF-CLIP) uses the holistic similarity score $r_\star = \cos(\mathbf{x}_\star, \mathbf{x})$.

## 4 Experiments

### 4.1 Dataset and Preprocessing

We explore two variants of the CLIP model, corresponding to either ResNet-50 ($\mathbf{x} \in \mathbb{R}^{1024}$) or ViT-B/32 ($\mathbf{x} \in \mathbb{R}^{512}$). ResNet-50 is a convolutional network, while ViT-B/32 incorporates vision transformers. Except for zero-shot concept classification, where the network's original embeddings are used, the image embeddings are first normalized to be unit norm, then re-centered using a pre-computed mean from Bhalla et al. (2024) based on the gap between the distributions of image and text embeddings of CLIP (Liang et al., 2022), before being re-normalized to unit norm again.

In addition to the vision-language model, we also explore TiTok-L-32 (Yu et al., 2024), a vision transformer-based autoencoder that compresses an input image into 32 pre-quantized tokens of dimension $d$, $\phi(X) \in \mathbb{R}^{32 \times d}$. We vectorize the tokens to get the embedding $\mathbf{x} \in \mathbb{R}^{384}$ with $d = 12$. Instead of normalizing across the whole embedding, we normalize each token separately to preserve the information each contains. Since there is no paired text encoder, no mean is removed from the embeddings. Finally, we explore DINOv2 ViT-B/14 (Jose et al., 2024), a self-supervised distilled image feature extraction model built on a vision transformer backbone. The embeddings are vectors $\mathbf{x} \in \mathbb{R}^{768}$, which we also normalize to unit norm without mean removal.

We used the MIRFlickr25K (Huiskes & Lew, 2008) and MS COCO (Lin et al., 2014) datasets for subsequent experiments. MIRFlickr25K contains $N = 25,000$ images with $S = 11$ general

concepts—6 concepts have finer-grained labels. MS COCO contains $N = 118,000$ images, 80 finer-grained labels under $S = 12$ general concepts. We removed images with all-zero labels from both datasets. We randomly selected 2,000 images from MIRFlickr25K and 20,000 images from MS COCO as the training set. A validation set of 500 images and a query set of 1,000 images were used to measure retrieval performance. The remaining images were used as candidates for retrieval.

We evaluate SLiCS and other baselines for the task of concept-filtered retrieval. In particular, we use unfiltered retrieval with CLIP (UF-CLIP), DINO (UF-DINO, TiTok (UF-TiTok), and SpLiCE (Bhalla et al., 2024). SpLiCE can either be used as an unfiltered retrieval method by reconstructing the image embedding based on the decomposed sparse coefficient, or as a filtered retrieval method by partially reconstructing the embedding to yield concept-specific components after the initial sparse coefficients decomposition, if a group structure of the word tokens is provided (F-SpLiCE). For F-SpLiCE, we group the tokens by assigning each to the closest centered concept word token $\mathbf{w}_i$ in terms of cosine similarity.

For a given concept $j$ (general label) and a query where the concept is present, the task is to retrieve images where $j$ is also present. A finer-grained concept-filtered retrieval score is also computed, where it is required to retrieve images that not only contain $j$ but also contain the same finer-grained concept (sub-label) of the query.[4] The retrieval is quantitatively measured by mean average precision of the top-20 retrieved images (mAP@20). It should be noted that for different $j$, the same query image may appear as query for different concepts. We select the hyperparameter $d_0$ in supervised SLiCS by comparing the finer-grained concept-filtered retrieval performance on a validation set. The optimal $d_0$ and the performances of general label retrieval and sub-label retrieval on different embeddings are shown in Fig. 5 in the Appendix.

For unsupervised SLiCS, $\tilde{S}$ is estimated by calculating the average number of concepts per sample across the whole dataset. The average number of concepts $\bar{S}$ in MIRFlickr25K and MS COCO is 2.467 and 2.303. Hence, we pick $\tilde{S} = 2$ for both datasets. We use the same $d_0$ that was selected for supervised SLiCS for a fair comparison.

In both supervised and unsupervised SLiCS, (S-SLiCS and U-SLiCS), we apply standard dictionary training on MIRFlickr25K, while applying the mini-batch training on MS COCO with a batch size of 2,000. Both are trained for 10 iterations. As an internal baseline, we use the initialization step of supervised SLiCS to create a supervised SVD-based dictionary (S-SVD).

### 4.2 Interpretation of the Concept Dictionary

The word-embedding based interpretation of concept dictionaries for CLIP embeddings are given in Table 3 in the appendix. Interpretation of atoms by prototypical images are in Fig. 20 and Fig. 21 in the appendix. The visualization in Fig. 2 (using image-to-prompt as described in Appendix A.2.4) shows how information is encoded in a subspace by the relative contribution of two atoms associated to the "animal" subspace. Increasing the number of atoms per concept is important to capture

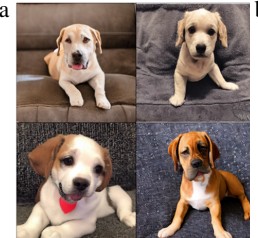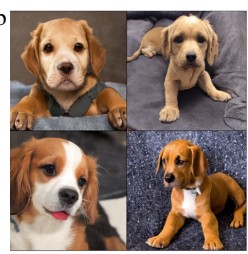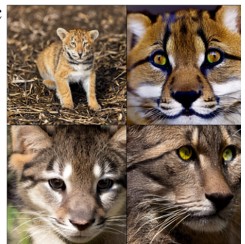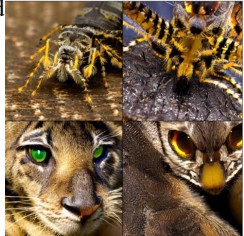

Figure 2: Generated images from image-to-prompt (Ding et al., 2023) applied to disentanglement in terms of two atoms $\mathbf{b}_1, \mathbf{b}_2$ from the "animal" subspace. (a) $0.25\mathbf{b}_1 + 0.75\mathbf{b}_2$. (b) $0.5\mathbf{b}_1 + 0.5\mathbf{b}_2$. (c) $0.75\mathbf{b}_1 + 0.25\mathbf{b}_2$. (d) $\mathbf{b}_1$. As the coefficients change, the content transitions from dog, associated to $\mathbf{b}_2$, to feline and spider-like animals with more intense colors and striping, associated to $\mathbf{b}_1$.

---

[4]For example, if the query contains "dog", but $j$ is coarse grained, such that it denotes "animal", then in normal concept-filtered retrieval, the relevant images are ones that also have "animal", while in finer-grained retrieval, a returned image is only considered relevant when the image contains "dog".

the diversity associated with each concept, but, with an increasing number of atoms, the cones of different concepts become closer (see Fig. 12 in the appendix). Within each concept-specific subspace, we can use finer-grained concepts not known during training to see how organized or structured it is. That is, during dictionary learning no sub-level information is given, but is only used in assessment. A visualization of the image embeddings and disentangled components by t-SNE is shown in Fig. 3. With an increasing $d_0$, the clusters representing finer-grained concepts first become separated and then merge together. This is confirmed by Fig. 13 and Fig. 14 in the appendix that measure the divergence between concepts and sub-concepts as a function of the number of atoms. Additional results in the appendix show that the atoms within a concept co-occur even when

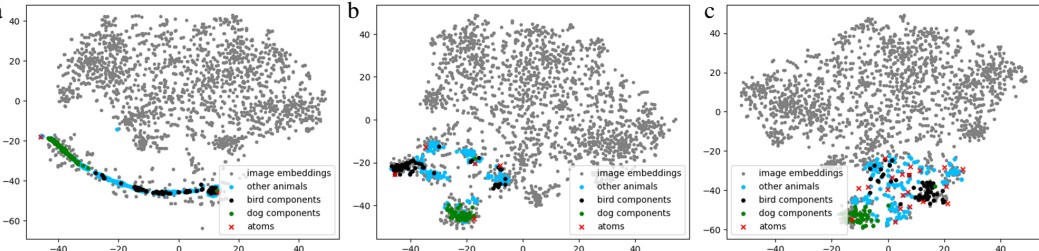

Figure 3: Visualization of disentangled subspaces by supervised SLiCS, with points corresponding to images with the "animal" concept colored using partial sub-label information. (a) $d_0 = 2$ shows the subspace curves between two atoms. (b) $d_0 = 5$ shows cluster structure within the subspace with the animal components that have "dog" or "bird" mostly within their own clusters. (c) $d_0 = 20$ shows less clustering as more atoms allows a varied landscape of components, but "bird" and "dog" remain grouped.

group structure is not imposed and the atoms within a concept are nearly orthogonal (Fig. 7), the coefficients often have long-tailed distributions (Fig. 15), and coefficients correlate within a concept (Fig. 16), i.e. the atoms are not independent components.

## 4.3 FILTERED AND UNFILTERED RETRIEVAL

We present the quantitative results in Table 1 (CLIP embeddings). S-SLiCS models perform the best on all but one case. The proposed initialization S-SVD is best on that case and second best on the remaining cases. The direct comparison between S-SLiCS and S-SVD shows the iterations of the dictionary learning algorithm consistently improve general label retrieval (increased mAP@20 by 0.011, 0.015, 0.014, and 0.024 for the 4 cases) and usually improve sub-label retrieval (differences of 0.021, -0.001, 0.043, and 0.066). Compared to unfiltered retrieval, S-SLiCS greatly improves general retrieval (0.152, 0.167, 0.14, 0.138) and moderately improves sub-label retrieval (0.056, 0.056, 0.054, 0.071). Sub-label retrieval is more difficult than general label retrieval, but SLiCS shows the potential to improve fine-grained image retrieval even when only general labels are given.

Table 1: Quantitative results (mAP@20) for CLIP embeddings. **Gen.** denotes general label retrieval score; **Sub.** denotes sub-label retrieval score. The best performance across each row is marked in bold and second best is underlined with ties for ±0.003.

| Type | | S-SVD | S-SLiCS | U-SLiCS | UF-CLIP | F-SpLiCE | SpLiCE |
|------|------|-------|---------|---------|---------|----------|--------|
| MIR/RN | Gen. | 0.864 | **0.875** | 0.832 | 0.723 | 0.653 | 0.646 |
| | Sub. | 0.735 | **0.756** | 0.726 | 0.700 | 0.565 | 0.578 |
| MIR/ViT | Gen. | 0.880 | **0.895** | 0.862 | 0.728 | 0.702 | 0.669 |
| | Sub. | 0.757 | **0.756** | 0.729 | 0.700 | 0.607 | 0.612 |
| COCO/RN | Gen. | 0.908 | **0.922 ± 0.001**[a] | 0.882 ± 0.002 | 0.782 | 0.628 | 0.707 |
| | Sub. | 0.674 | **0.717 ± 0.003** | 0.651 ± 0.003 | 0.663 | 0.496 | 0.584 |
| COCO/ViT | Gen. | 0.905 | **0.929 ± 0.002** | 0.887 ± 0.001 | 0.791 | 0.684 | 0.737 |
| | Sub. | 0.676 | **0.742 ± 0.002** | 0.673 ±0.001 | 0.671 | 0.539 | 0.627 |

[a]The mean and standard deviation are calculated across five runs for mini-batch ordering.

Except for COCO/RN sub-label, the unsupervised SLiCS (U-SLiCS) models perform third best. U-SLiCS performs well in general label retrieval (outperforming UF-CLIP by 0.109, 0.134, 0.083, and 0.08 across the 4 cases). For sub-label retrieval, U-SLiCS outperforms UF-CLIP on MIR-Flickr25K by 0.026 and 0.029, but matches it on MS COCO. These results highlight the utility using pseudo-labels from zero-shot classification to enable unsupervised SLiCS for concept-filtered image retrieval. Table 7 in the appendix shows per concept level retrieval for the ViT-B/32 embeddings on MS COCO.Table 6 in the appendix shows that dynamic selection yields slightly higher general label performance but lower sub-label performance.

Finally, we note that unfiltered CLIP always outperforms either of the SpLiCE baselines. SpLiCE's unfiltered approximation does not achieve disentanglement, and the post-hoc grouping of atoms in F-SpLiCE based on similarity to the concept text embedding does not help.

### 4.3.1 TiTok Embeddings

To show the wider applicability of SLiCS, we apply it to the latent "token" embeddings from TiTok. In Table 2 (first three numeric columns), we present the results of supervised SLiCS and unfiltered TiTok embeddings. Quantitative results show that SLiCS provides much higher precision compared to unfiltered retrieval with increases in mAP@20 of 0.21 and 0.23 for general labels, and increases of 0.08 and 0.04 for sub-labels. The supervised SVD initialization actually outperforms the learned dictionary on 3 of the 4 cases. This indicates that for the TiTok embedding space, which was not trained with InfoNCE objective like CLIP, the non-negativity constraint which yields positive cones may not be necessary compared to linear subspaces. Nonetheless, these results highlight the importance of using disentanglement to provide meaningful retrieval. The fact that TiTok embeddings capture spatial arrangement (as they are trained with reconstruction-based loses) in addition to semantics likely contributes to the lower retrieval performance (as similar visual arrangement of scenes may dominate the similarity) and the large gains from concept filtering.

### 4.3.2 DINO Embeddings

We also apply SLiCS to the embeddings of the self-supervised DINOv2 ViT-B/14 model (Jose et al., 2024), which creates embeddings of images using a vision transformer backbone. The results of supervised SLiCS and unfiltered DINO embeddings are presented in Table 2, with more details in Table 7 in the appendix. The improvements S-SLiCS over unfiltered retrieval show the efficacy

Table 2: Quantitative results (mAP@20) for TiTok (-T) and DINO (-D) embeddings. The best performance per dataset-embedding pair is marked in bold and second best is underlined.

|  |  | S-SVD-T | S-SLiCS-T | UF-T | S-SVD-D | S-SLiCS-D | UF-D |
|---|---|---|---|---|---|---|---|
| MIR | Gen. | 0.724 | **0.738** | 0.523 | 0.912 | **0.943** | 0.726 |
|  | Sub. | **0.516** | 0.500 | 0.426 | 0.737 | **0.758** | 0.700 |
| COCO | Gen. | **0.823** | $0.795 \pm 0.004$ | 0.564 | 0.909 | **0.984 $\pm$ 0.001** | 0.793 |
|  | Sub. | **0.421** | $0.382 \pm 0.005$ | 0.341 | 0.728 | **0.785 $\pm$ 0.004** | 0.698 |

of disentanglement, and the improvements on S-SVD initialization shows the efficacy of our dictionary learning algorithm.[5] It should be noted that mAP@20 with the disentanglement of DINO embeddings is better than that with the disentanglement of CLIP embeddings. We note that DINO's self-supervision tries to extract latent embeddings that are invariant to augmentations. The resulting embeddings may capture various aspects that differentiate the whole contents. In comparison, the CLIP model can be seen as weakly supervised, as it is trained to capture the semantic information in the image to match the paired caption. The way CLIP encodes the images is restricted by semantics with the guidance of captions, the quality of which restrict its abilities (Gurung et al., 2025), while DINO tries to extract anything that is unique about an image. The qualitative results shown in Fig. 4 corroborates the disentanglement of the DINO embeddings. Top-5 retrieval results are in Fig. 18 and Fig. 19 in the appendix. Concept-filtered retrieval accurately retrieves images with objects aligned

---

[5]Ablation study for the non-negative constraint are shown in Table 4 in the appendix along with results using a variable number of atoms proportional to the concept frequency or chosen to ensure a common proportion variance explained in Table 6. The results of the ablation study show removing non-negativity decreases the performance of SLiCS. The study on choice on number of atoms shows robustness in performance across different schemes.

Figure 4: Qualitative results for DINOv2 ViT-B/14 embeddings (MS COCO). The concepts per image are sorted in descending cosine similarity between the disentangled components and the query.

with the concept in the query: the first query yields cow ("animal") and kids ("person"), the second query yields black cat ("animal") and suitcases ("accessory"), the third query yields man with a pair of sunglasses ("person") and a cellphone being used ("electronic"). Qualitative results for the other embeddings are shown in Appendix A.2.3. We also show the visualization of the disentangled components and the concept-specific cones in Appendix A.2.4 and Appendix A.2.5.

## 5 DISCUSSION

SLiCS is motivated by image retrieval from large image datasets consisting of complex scenes. Another application is to cluster the components of each concept to find variation of a concept across scenes. Clustering of a concept's disentangled concepts in the latent space is an alternative to how image segmentation is used by Ghorbani et al. (2019) to discover concepts associated with a class. Without disentanglement or input space segmentation, clusters of images with a concept without disentanglement may be less informative due to the other active concepts. In the case of vision-language co-embeddings, descriptions of these clusters could be found automatically by the text embeddings best fit by the cluster. Future work could also investigate the use of concept-component based editing (removal of one or all but one component), especially if the embeddings are used in downstream tasks (Koh et al., 2020).

SLiCS also provides a lens to understand the composition of latent embeddings, even without text co-embeddings. As described in Appendix A.2.5 and shown in Figure 11, diffusion posterior sampling from cones allows an understanding of each cone. This approach could be used with concepts learned through completely unsupervised dictionary learning (Szabó et al., 2011) or automatically for classification tasks (Ghorbani et al., 2019; Yeh et al., 2020). SLiCS achieves disentanglement of concept subspaces by modeling their co-activity. The individual atoms associated to a concept define the boundaries of the cone, as in archetypal analysis (Cutler & Breiman, 1994; Mørup & Hansen, 2012). Future work can also investigate more attribute-level concepts (Koh et al., 2020) and hierarchically organized concepts with hierarchical groupings of dictionaries, in comparison to the partitions explored here.

While we focused a single vector embedding per image (the class token in CLIP and DINO), future work can apply SLiCS to spatially arranged vectors (reshaped tensors in convolutional neural networks) as used to produce salience maps in related work (Zhang et al., 2021; Fel et al., 2023).

## 6 CONCLUSION

SLiCS enables the decomposition of image embeddings (we explored CLIP, TiTok, and DINO) into meaningful concept subspaces. The methods uses dictionary learning with supervision from labels or pseudo-labels from zero-shot classification based on minimal guidance from the text embeddings of a known set of concepts. The immediate benefit of the decomposition is increased effectiveness of concept-filtered image retrieval and concept-specific conditional generation. For CLIP, SLiCS also enables interpretable descriptions in both supervised and unsupervised cases. We found that DINOv2 (Jose et al., 2024) that uses self-supervised training to capture rich information from images offered the best concept-filtered retrieval performance. Its unfiltered retrieval performance matched CLIP's performance, and applying SLiCS to DINO embeddings yielded precise concept-filtered retrieval (0.943 mAP@20 on MIRFlikr25K and 0.984 on MS COCO). Our concept-level disentanglement indicates that non-negative, group-sparse linear synthesis model is a promising and viable perspective to understand the latent embeddings of vision-language models and other deep neural networks.

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

# A  APPENDIX

## A.1  SUPERVISED SLiCS DICTIONARY LEARNING ALGORITHM

---

**Algorithm 1** Supervised SLiCS dictionary learning

---

**Require:** Embedding matrix $\mathbf{X} \in \mathbb{R}^{d \times N}$, cardinality of each dictionary $M_1, \ldots, M_S$ with $M = \sum_{j=1}^{S} M_j$, concept indices from labels, $\mathcal{I}_j \subset \{1, \ldots, N\}, j \in \{1, \ldots, S\}$, max iterations $T$

**Ensure:** Dictionary $\mathbf{B} = [\mathbf{B}_j]_{j=1}^{S} = [\mathbf{b}_m]_{m=1}^{M}$

    **for** $j = 1$ to $S$ **do**

        Initialize $\mathbf{B}_j \leftarrow [p_k^* \mathbf{u}_k^L]_{k=1}^{M_j}$ from the rank-$M_j$ truncated SVD of $[\mathbf{x}_i]_{i \in \mathcal{I}_j} \approx \sum_{k=1}^{M_j} \sigma_k \mathbf{u}_k^L \mathbf{u}_k^{R\top}$

        with optimal signs $p_k^* = \begin{cases} 1 & \|\max(0, \mathbf{u}_k^R)\|_2 \geq \|\max(0, -\mathbf{u}_k^R)\|_2 \\ -1 & \text{otherwise} \end{cases}$

    **end for**

    **for** $t = 1$ to $T$ **do**

        **for** $i = 1$ to $N$ **do**

            Solve $\boldsymbol{\alpha}^i = [\boldsymbol{\alpha}_j^i]_{j=1}^{S}$ using NNLS on supported concepts, $\boldsymbol{\alpha}_j^i = \mathbf{0}$ if $i \notin \mathcal{I}_j$, as in equation 5

        **end for**

        $\mathbf{A} \leftarrow [\boldsymbol{\alpha}^i]_{i=1}^{N}$

        **for** $m = 1$ to $M$ **do**

            Find the support indices $\mathcal{L}_m \leftarrow \{l \in \{1, \ldots, M\} : A_{ml} \neq 0\}$ of the $m$th row of $\mathbf{A}$

            Calculate the residual error without the $m$th atom $\mathbf{E} \leftarrow [\mathbf{x}_l - \sum_{k \neq m} \mathbf{b}_k A_{kl}]_{l \in \mathcal{L}_m}$

            Obtain rank-1 truncated SVD of residual error $\mathbf{E} \approx \tilde{\mathbf{b}}_m \tilde{\boldsymbol{\beta}}$,

            Compute the optimal sign $p^\star = \begin{cases} 1 & \|\max(0, \tilde{\boldsymbol{\beta}})\|_2 \geq \|\max(0, -\tilde{\boldsymbol{\beta}})\|_2 \\ -1 & \text{otherwise} \end{cases}$

            Update the atom $\mathbf{b}_m \leftarrow p^\star \tilde{\mathbf{b}}_m$

            Update the active coefficients $\mathbf{A}_{m\mathcal{L}_m} \leftarrow \max(0, p^\star \tilde{\boldsymbol{\beta}}) = \arg\min_{\boldsymbol{\beta} \geq \mathbf{0}} \|\mathbf{E} - \mathbf{b}_m \boldsymbol{\beta}\|_F$

        **end for**

    **end for**

---

### A.1.1  OPTIMAL SIGN FOR ATOM UPDATE

The optimality of the majority rule follows from the fact that the non-negative least squares problem can be decomposed into solving one entry of $\mathbf{A}_{m\mathcal{L}_m} = [A_{ml}]_{l \in \mathcal{L}_m}$ at a time. Since the solution of non-negative linear regression is given by the active set method (Bro & De Jong, 1997), the entry is either included in the active set if it is non-negative, or not included if it is negative. Hence, the optimal non-negative $\beta^\star$ is obtained by setting the negative coefficient to zero. Additionally, the rank-1 SVD already guarantees the optimal result of ordinary linear regression $\tilde{\boldsymbol{\beta}}$, which allows us to prove that the majority sign rule reduces the error. We begin by decomposing the norm of the error of the rank-1 approximation of the residual,

$$
\begin{aligned}
\|\mathbf{E} - \tilde{\mathbf{b}}_m \tilde{\boldsymbol{\beta}}\|_F^2 &= \sum_{l \in \mathcal{L}_m} \|\mathbf{e}_l - \tilde{\mathbf{b}}_m \tilde{\beta}_l\|_2^2 = \sum_{l \in \mathcal{L}_m} \left( \|\mathbf{e}_l\|_2^2 + \underbrace{\|\tilde{\mathbf{b}}_m \tilde{\beta}_l\|_2^2}_{\tilde{\beta}_l^2} - 2 \underbrace{\tilde{\beta}_l \mathbf{e}_l^\top \tilde{\mathbf{b}}_m}_{\tilde{\beta}_l^2} \right) \\
&= \sum_{l \in \mathcal{L}_m} \left( \|\mathbf{e}_l\|_2^2 - \tilde{\beta}_l^2 \right) = \sum_{l \in \mathcal{L}_m} \left( \|\mathbf{e}_l\|_2^2 - \max(0, \tilde{\beta}_l)^2 - \max(0, -\tilde{\beta}_l)^2 \right) \\
&= \|\mathbf{E}\|_F^2 - \|\max(0, \tilde{\boldsymbol{\beta}})\|_2^2 - \|\max(0, -\tilde{\boldsymbol{\beta}})\|_2^2,
\end{aligned}
\tag{7}
$$

where the simplifications follow from the fact that the optimal least-square coefficient $\tilde{\beta}_l = \mathbf{e}_l^\top \tilde{\mathbf{b}}_m$ is simply the inner-product with the unit vector $\tilde{\mathbf{b}}_m$. This can also be seen from the SVD, since

$$\left[ \tilde{\mathbf{b}}_m^\top \mathbf{e}_l \right]_{l \in \mathcal{L}_m} = \tilde{\mathbf{b}}_m^\top \mathbf{E} = \tilde{\mathbf{b}}_m^\top \mathbf{U}^L \mathbf{\Sigma} \mathbf{U}^{R\top} = \sigma_1 \mathbf{u}_1^{R\top} = \tilde{\boldsymbol{\beta}} \tag{8}$$

$$\tilde{\mathbf{b}}_m^\top \mathbf{e}_l = \tilde{\beta}_l.$$

For arbitrary sign $p \in \{-1, 1\}$, and given the non-negative thresholding operator $(\cdot)_+ = \max(0, \cdot)$, we note that

$$\sum_{l \in \mathcal{L}_m} \min_{\beta \geq 0} \| \mathbf{e}_l - p\tilde{\mathbf{b}}_m \beta \|_2^2 = \sum_{l \in \mathcal{L}_m} \| \mathbf{e}_l - p\tilde{\mathbf{b}}_m (p\underbrace{\mathbf{e}_l^\top \tilde{\mathbf{b}}_m}_{\tilde{\beta}_l})_+ \|_2^2 = \| \mathbf{E} - p\tilde{\mathbf{b}}_m (p\tilde{\boldsymbol{\beta}})_+ \|_2^2, \tag{9}$$

since the optimal non-negative least-square coefficient is the non-negative projection of the optimal least squares component $\beta_l = \max(0, p\mathbf{e}_l^\top \tilde{\mathbf{b}}_m)$. To show the optimality of $p^*$ defined by the majority sign, the non-negative approximation can be expanded as in equation 7, yielding

$$\| \mathbf{E} - p\tilde{\mathbf{b}}_m (p\tilde{\boldsymbol{\beta}})_+ \|_F^2 = \sum_{l \in \mathcal{L}_m} \| \mathbf{e}_l - p\tilde{\mathbf{b}}_m (p\tilde{\beta}_l)_+ \|_2^2$$

$$= \sum_{l \in \mathcal{L}_m} \left( \| \mathbf{e}_l \|_2^2 + \underbrace{p^2}_{1} \underbrace{\| \tilde{\mathbf{b}}_m (p\tilde{\beta}_l)_+ \|_2^2}_{(p\tilde{\beta}_l)_+^2 \| \tilde{\mathbf{b}}_m \|_2^2 = (p\tilde{\beta}_l)_+^2} -2 \underbrace{p(p\tilde{\beta}_l)_+ \tilde{\mathbf{b}}_m^\top \mathbf{e}_l}_{p(p\tilde{\beta}_l)_+ \tilde{\beta}_l = (p\tilde{\beta}_l)_+^2} \right) \tag{10}$$

$$= \sum_{l \in \mathcal{L}_m} \left( \| \mathbf{e}_l \|_2^2 - (p\tilde{\beta}_l)_+^2 \right)$$

$$= \| \mathbf{E} \|_F^2 - \| (p\tilde{\boldsymbol{\beta}})_+ \|_2^2$$

$$\geq \| \mathbf{E} \|_F^2 - \max(\| (\tilde{\boldsymbol{\beta}})_+ \|_2^2, \| (-\tilde{\boldsymbol{\beta}})_+ \|_2^2) = \| \mathbf{E} - p^\star \tilde{\mathbf{b}}_m (p^\star \tilde{\boldsymbol{\beta}})_+ \|_F^2,$$

where the following equality was used, which holds for any scalar $\beta$

$$p(p\beta)_+ \beta = p\beta(p\beta)_+ = \begin{cases} (p\beta)^2, & (p\beta)_+ \geq 0 \\ 0, & (p\beta)_+ < 0 \end{cases} \tag{11}$$

$$= (p\beta)_+^2.$$

This shows that $p^\star$ computed by the majority rule is always the optimal sign.

### A.1.2 CONVERGENCE

The first step in each iteration, the coefficient update, always reduces the reconstruction error as non-negative linear regression is a convex minimization problem with a unique global minimum. The update of each atom (and coefficients) can be made contingent on improving the approximation, i.e., the atom update can be skipped if the newly found coefficients and the new atom with optimal sign do not provide a better rank-1 approximation compared to the original atom and previous coefficients. Mathematically, the update improves the approximation if

$$\| \mathbf{E} - p^\star \tilde{\mathbf{b}}_m (p^\star \tilde{\boldsymbol{\beta}})_+ \|_F^2 = \sum_{l \in \mathcal{L}_m} \| \mathbf{x}_l - \sum_{k \neq m} \mathbf{b}_k A_{kl} - p^* \tilde{\mathbf{b}}_m (p^* \tilde{\beta}_l)_+ \|_2^2 \leq \sum_{l \in \mathcal{L}_m} \| \mathbf{x}_l - \sum_k \mathbf{b}_k A_{kl} \|_2^2. \tag{12}$$

With this check, the algorithm is a cyclic minimizer (Stoica & Selen, 2004), and thus is guaranteed to converge since both stages reduce the error.

In practice, we let the algorithm proceed with an atom update without checking if it will increase the reconstruction error, as it may end up achieving a better local minimum.

Finally, we note that an alternative is to use block-coordinate descent on the atom and coefficient update, which will always reduce the error (see Section A.1.3); however, in simulations our simultaneous approach using SVD and thresholding with optimal sign leads to a better approximation.

### A.1.3 ALTERNATIVE UPDATE OF ATOMS

In K-SVD, using SVD to simultaneously update coefficients and atoms during the atom update step is optimal to reduce the norm of the error in the residual. However, with an additional non-negative constraint, it is not optimal, as the problem is a rank-1 semi-NMF problem (Gillis & Kumar, 2015). Rather than our heuristic, which uses the thresholded coefficients after identifying the optimal sign, an alternative is to perform block-coordinate descent by updating atom and coefficients separately. Given the previous coefficients, each entry of the atom can be optimized in parallel by ordinary least squares. Then, given updated atom, the non-negative linear regression can be used to obtain the updated coefficients, again in parallel:

$$\acute{\mathbf{b}}_m = \arg\min_{\boldsymbol{\gamma}} \|\mathbf{E} - \boldsymbol{\gamma}\boldsymbol{\beta}_0\|_F^2, \qquad \acute{\boldsymbol{\beta}} = (\breve{\boldsymbol{\beta}})_+, \quad \breve{\boldsymbol{\beta}} = \arg\min_{\boldsymbol{\beta}} \|\mathbf{E} - \acute{\mathbf{b}}_m\boldsymbol{\beta}\|_F^2, \qquad (13)$$

where $\boldsymbol{\beta}_0 = [A_{ml}]_{l \in \mathcal{L}_m}$ from the previous coefficient update. Both of these problems have closed-form solutions, $\acute{\mathbf{b}}_m = \mathbf{E}\boldsymbol{\beta}_0^\top(\boldsymbol{\beta}_0\boldsymbol{\beta}_0^\top)^{-1} = \dfrac{\mathbf{E}\boldsymbol{\beta}_0^\top}{\|\boldsymbol{\beta}_0\|^2}$ and $\breve{\boldsymbol{\beta}} = (\acute{\mathbf{b}}_m^\top\acute{\mathbf{b}}_m)^{-1}\acute{\mathbf{b}}_m^\top\mathbf{E} = \dfrac{\acute{\mathbf{b}}_m^\top\mathbf{E}}{\|\acute{\mathbf{b}}_m\|^2}$, respectively. With the inclusion of a normalizing step to enforce $\acute{\mathbf{b}}_m$ to have unit norm, the solutions are scaled yielding

$$\acute{\mathbf{b}}_m = \frac{\mathbf{E}\boldsymbol{\beta}_0^\top}{\|\mathbf{E}\boldsymbol{\beta}_0^\top\|}, \qquad \breve{\boldsymbol{\beta}} = (\acute{\mathbf{b}}_m^\top\acute{\mathbf{b}}_m)^{-1}\acute{\mathbf{b}}_m^\top\mathbf{E} = \frac{\boldsymbol{\beta}_0\mathbf{E}^\top\mathbf{E}}{\|\boldsymbol{\beta}_0\mathbf{E}^\top\|}. \qquad (14)$$

Following Eq. 10, the residual error after the update is

$$\|\mathbf{E} - \acute{\mathbf{b}}_m(\breve{\boldsymbol{\beta}})_+\|_F^2 = \|\mathbf{E}\|_F^2 - \|(\breve{\boldsymbol{\beta}})_+\|_2^2. \qquad (15)$$

The residual error can be lower bounded by upper bounding the second term as

$$\|(\breve{\boldsymbol{\beta}})_+\|_2^2 \leq \|\breve{\boldsymbol{\beta}}\|_2^2 = \frac{\|\boldsymbol{\beta}_0\mathbf{E}^\top\mathbf{E}\|_2^2}{\|\boldsymbol{\beta}_0\mathbf{E}^\top\|_2^2} \leq \left\|\frac{\boldsymbol{\beta}_0\mathbf{E}^\top}{\boldsymbol{\beta}_0\mathbf{E}^\top}\right\|_2^2 \cdot \|\mathbf{E}\|_{\mathrm{op}}^2 = \sigma_1^2 = \|\tilde{\boldsymbol{\beta}}\|_2^2. \qquad (16)$$

Although it is hard to compare between $\|(\breve{\boldsymbol{\beta}})_+\|$ and $\|(p^*\tilde{\boldsymbol{\beta}})_+\|$, we know from equation 16 that the total norm of $\breve{\boldsymbol{\beta}}$ is smaller than $\tilde{\boldsymbol{\beta}}$, making it less likely to reduce more residual error. Furthermore, it also shows that at the beginning phase of the iteration, where $\acute{\mathbf{b}}_m^\top$ is far apart from the space described by $\mathbf{E}$, the last inequality will be stronger. In fact, it is easy to show that when the true optimum is close, i.e., when $\mathbf{E}$ can be well-approximated by a rank-1 matrix, these two methods are equal. Assuming $\mathbf{E} = \sigma_1\mathbf{u}_1^L\mathbf{u}_1^{R\top}$, since $\acute{\mathbf{b}}_m^\top$ needs to lie in the space of $\mathbf{E}$, which only has one left singular vector, leading to $\acute{\mathbf{b}}_m^\top = \mathbf{u}_1^{L\top}$, then

$$\breve{\boldsymbol{\beta}} = \sigma_1\mathbf{u}_1^{L\top}\mathbf{u}_1^L\,\mathbf{u}_1^{R\top} = \sigma_1\mathbf{u}_1^{R\top} = \tilde{\boldsymbol{\beta}}. \qquad (17)$$

Considering that new coefficients can be written in terms of previous coefficients, given the initial update of non-negative set of coefficients, $\acute{\boldsymbol{\beta}}_1 = (\breve{\boldsymbol{\beta}})_+$, a heuristic approach is to iteratively update,

$$\acute{\boldsymbol{\beta}}_{t+1} = \left(\frac{\acute{\boldsymbol{\beta}}_t\mathbf{E}^\top\mathbf{E}}{\|\acute{\boldsymbol{\beta}}_t\mathbf{E}^\top\|}\right)_+, \qquad (18)$$

for $t \in \{1, \ldots, T\}$ as a modified power method.

## A.2 ADDITIONAL RESULTS

### A.2.1 INTERPRETATION VIA WORD CAPTIONS

A set of words are selected to interpret the subspaces learned by SLiCS. We consider the single English word token from the captions of LAION-400m dataset, and follow Bhalla et al. (2024) to filter out any NSFW words and pick the most frequent 10,000 words. The text embeddings are normalized, mean-centered, and then re-normalized. Top 5 words that have the smallest non-negative linear reconstruction error using the concept dictionary $\mathbf{B}_j$ are selected to describe the subspace $\mathcal{V}_j$ since they are the closest to the subspace in the embedding space.

The word captions of supervised and unsupervised SLiCS with ViT-B/32 embeddings are presented in Table 3. As shown in the tables, both supervised and unsupervised subspace disentanglement show semantic consistency. However, unsupervised word caption lists seem less disentangled. For example, "aircraft", "plane", and "airplanes" are assigned in "electronic" and "appliance", possibly due to the inaccurate pseudo-labels due to photos within airplane cockpits or cabins containing electronics or appliances. As another example, "workspace" is shared by "kitchen" and "furniture".

Table 3: Subspace captions of MS COCO on CLIP ViT-B/32 embeddings. "Subspace" column shows the ground truth concept label. Words are ordered by the ascending reconstruction error.

| Subspace | Supervised-SLiCS word caption | Unsupervised-SLiCS word caption |
|---|---|---|
| person | kid, kids, queen, granddaughter, daughter | granddaughter, kid, grandson, daughter, himself |
| vehicle | bike, bus, boats, boat, aircraft | transport, transportation, touring, truck, motorbike |
| outdoor | bench, benches, downtown, streets, hometown | downtown, snow, streets, snowfall, tourists |
| animal | dogs, pups, horses, cows, puppies | horse, cow, horses, bear, dogs |
| accessory | travelers, luggage, baggage, travel, travelling | packing, luggage, baggage, suitcase, travelling |
| sports | batting, flying, hikers, baseball, skiing | kids, surfers, children, childrens, baseball |
| kitchen | beverages, drinks, breakfast, alcohol, beverage | kitchen, kitchens, bathroom, cooking, workspace |
| food | veggies, vegetables, pizza, cake, sandwich | meal, desserts, breakfast, healthy, cake |
| furniture | room, bedroom, bedrooms, bed, toilet | furniture, lounge, beds, packing, workspace |
| electronic | office, workspace, television, tv, phone | phone, airplanes, midnight, plane, nights |
| appliance | kitchen, kitchens, bathroom, sinks, refrigerator | aircraft, toilet, plane, landed, locomotive |
| indoor | library, books, bookshelf, libraries, decor | hall, basement, museum, warehouse, stores |

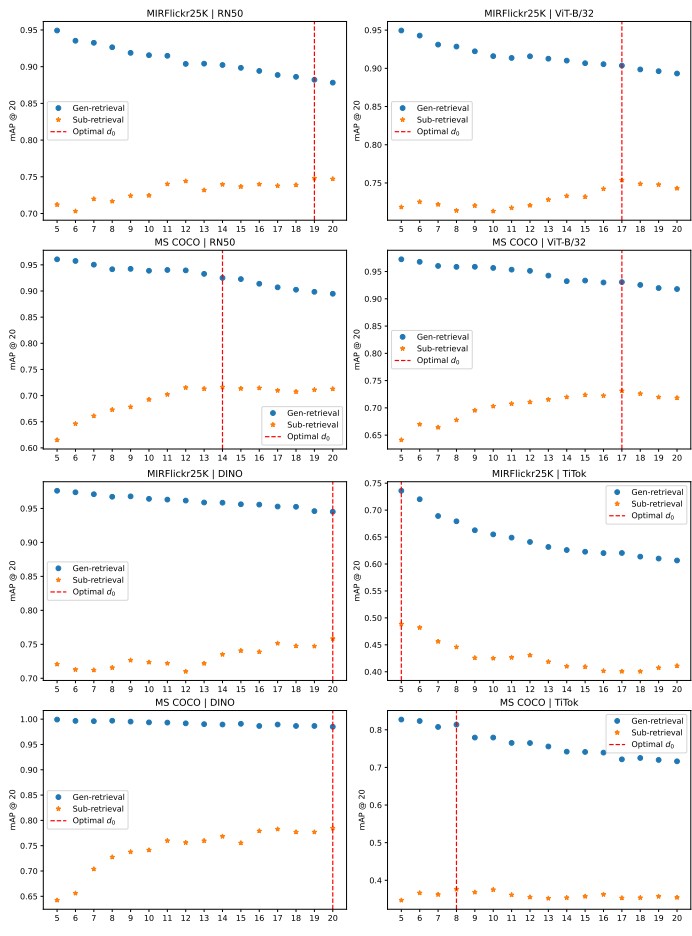

Figure 5: mAP@20 of general label retrieval and sub-label retrieval with various $d_0$ on the validation set of different embeddings. $d_0$ is searched from 5 to 20. The chosen $d_0$ is denoted by the vertical line.

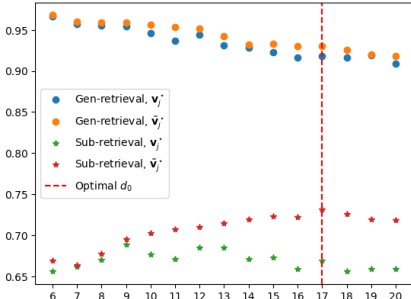

Figure 6: mAP@20 of general label retrieval and sub-label retrieval with various $d_0$ on ViT-B/32 embeddings of MS COCO using $\mathbf{v}_j^\star$ and $\tilde{\mathbf{v}}_j^\star$. The chosen $d_0$ is denoted by the vertical line.

### A.2.2 ATOM CO-OCCURRENCE VERSUS ATOM SIMILARITY

To emphasize that the group structure creates atoms that work together, coefficients are estimated without using the concept labels using greedy sparse coding (Orthogonal Matching Pursuit (Mallat & Zhang, 1993; Rubinstein et al., 2008) with non-negative constraint) across the validation set examples. Then the co-occurrence between atoms is calculated from the frequency that one atom co-occurs with another based on both having non-zero coefficients. Fig. 7 shows that the group structure is evident from the co-occurrence matrix, whereas the atoms within a concept group are nearly orthogonal (as seen by the cosine similarity). We also confirmed that there is no redundancy of atoms for the same concept in that none are interior to the convex hull for all the tested values of $d_0$. During training, atoms associated to one concept are always allowed to co-occur compared

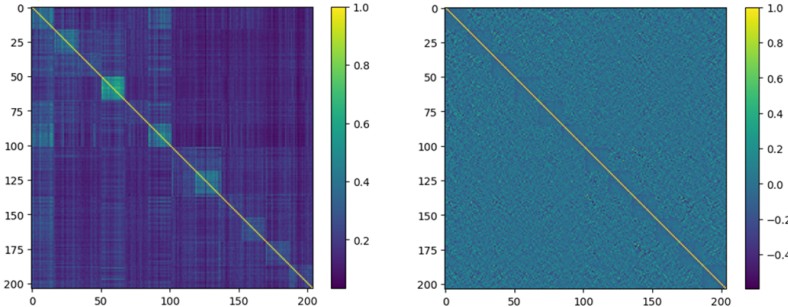

Figure 7: Relationship between the atoms across the concept dictionaries and their coefficients in supervised SLiCS. Dimensionality chosen by retrieval performance on validation set (see Section 4.3). (Left) the co-occurrence matrix of active atoms for images in the validation set. Each row is normalized separately so that the diagonal entries are 1. Within each group the co-occurrence rate is generally higher than between groups. (Right) The cosine similarity matrix of the atoms in the same dictionary. The within-group similarity $(-0.056 \pm 0.062)$ is slightly lower than cross-group similarity $(0.000 \pm 0.134)$.

to atoms from other concepts; thus, they tend to be nearly orthogonal to better capture the diversity within a concept.

### A.2.3 QUALITATIVE RETRIEVAL RESULTS

The qualitative results on MS COCO queries of ViT-B/32 embeddings are shown in Fig. 8. Although UF-CLIP and F-SpLiCE are able to retrieve similar images with query, they are not able to separate apart different concepts to enable a more specific retrieval as S-SLiCS. More specifically, S-SLiCS accurately retrieves a child with clothes of similar colors (first query) and a cat of similar color pattern (second query). U-SLiCS, on the other hand, retrieves a cat of disparate colors and a child less similar to the query, despite both correct finer-grained concepts.

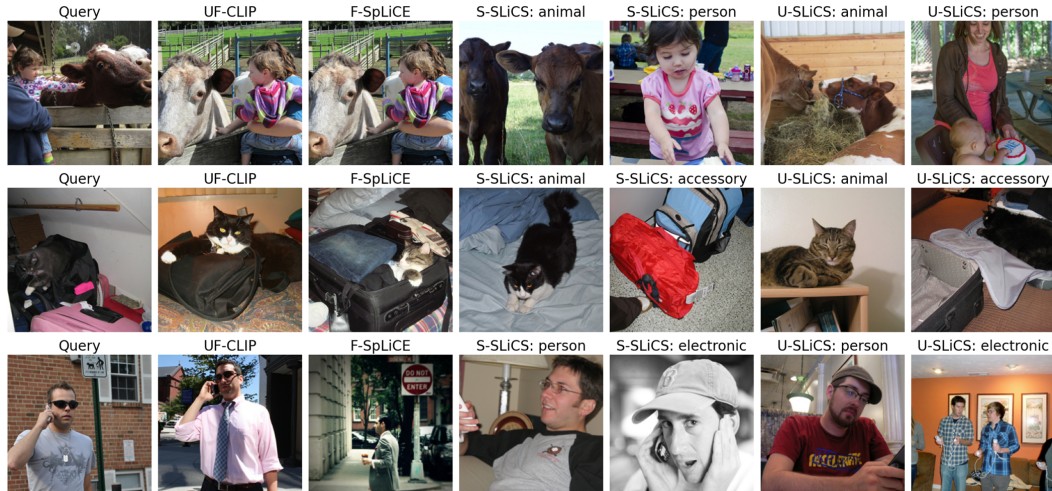

Figure 8: Qualitative results for CLIP ViT-B/32 embeddings (MS COCO). The concepts per image are sorted in descending cosine similarity between the disentangled components and the query. S-SLiCS shows effective decomposition of the query into relevant concepts. Unfiltered retrieval methods can only retrieve images based on the whole scene, for example, in the first row, "cow" and "person" need to co-occur with similar relative position. In contrast, SLiCS applies concept-filtered retrieval obtaining images with a similar cow and a similar person separately.

The qualitative results on MS COCO queries of TiTok-L-32 embeddings are shown in Fig. 9. We also show the reconstruction of the queries using TiTok encoder to display the distortions brought by the compressive encoding. Unfiltered TiTok retrieval yields images similar to queries in terms of low-level spatial features but not in terms of semantic contents, which reveals the limitations of using cosine similarity as the metrics on other models than CLIP. Specifically, in the second query, the fact that supervised SLiCS fails to retrieve the correct animal may be due to the information is distorted during encoding, as the reconstruction image yields a dog rather than a cat.

### A.2.4 Image-to-prompt Visualization

Stable diffusion models often use a CLIP text embedding as the prompt to handle image generation. Ding et al. (2023) proposes a method to directly use an image embedding as the prompt. However, it requires that the image and text embeddings lie in the same contrastively learned latent space. Since the training of the stable diffusion model (Rombach et al., 2022) is conditioned on CLIP ViT-L/14, it is necessary to align the image embedding spaces first. We formulate the space alignment problem as the well-known orthogonal Procrustes problem (Schönemann, 1966), which aims to find an orthogonal matrix that minimizes the mean-squared error as a regression problem between the embeddings. It has a closed-form solution in terms of the singular value decomposition of the product of the two embedding matrices. We choose Flickr30K dataset (Young et al., 2014), which contains 30,000 images with captions and is commonly used in cross-modal learning tasks, to align the spaces. The resulting orthogonal matrix is applied to $\tilde{\mathbf{v}}_j^\star$ as the prompt for the $j$th concept.

The image-to-prompt realizations created using the concept-specific components of the query are shown in Fig. 10. While the TiTok decoder could be applied to the components of the query, as it was trained on unit-norm tokens, passing in tokens with less than unit-norm fails and quantization of the component-tokens with unit-norm codebook distorts the component and creates unnatural reconstructions. Like the compression of the components, this is left for future work.

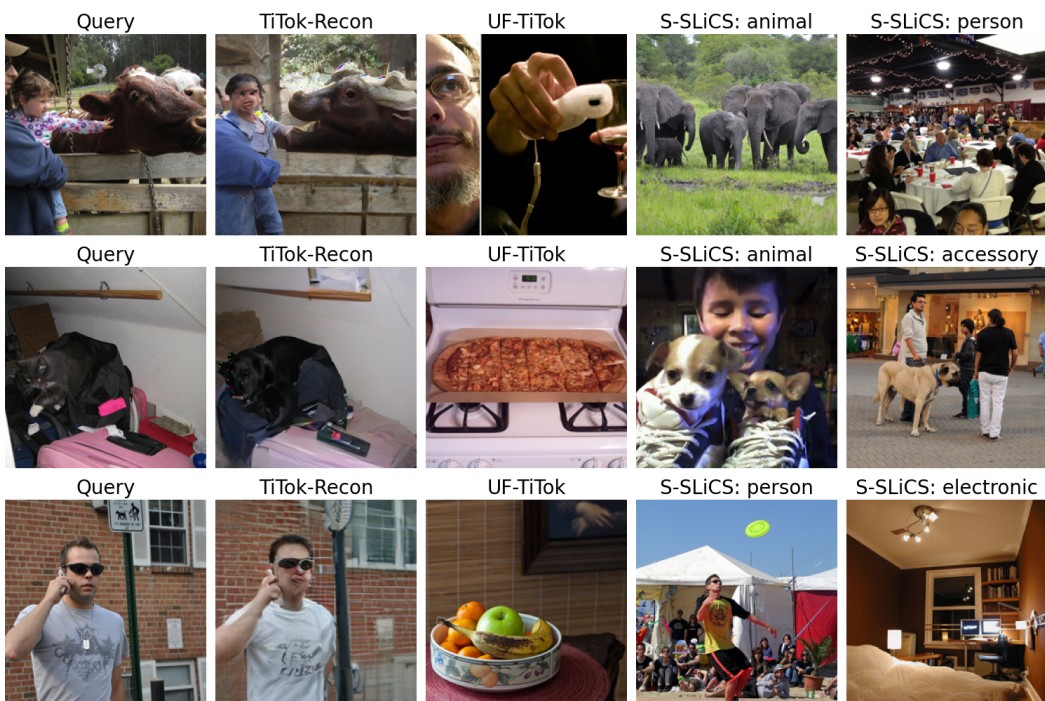

Figure 9: Qualitative results for TiTok-L-32 embeddings (MS COCO). The concepts per image are sorted in descending cosine similarity between the disentangled components and the query. **TiTok-Recon** column shows the TiTok reconstruction of the query images from the tokens, which bears distortions of varying degrees.

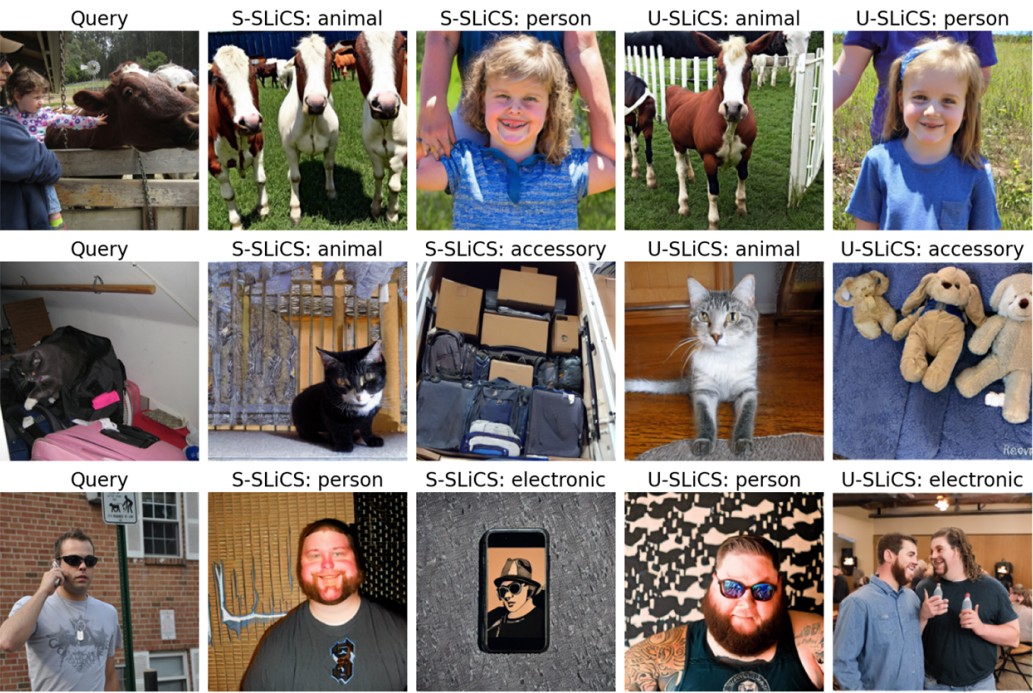

Figure 10: Image-to-prompt visualization for SLiCS model. The generated images in each row correspond to the components of each query in Fig. 8.

### A.2.5 DIFFUSION POSTERIOR SAMPLING

Diffusion posterior sampling (DPS) Chung et al. (2022) is a method to sample from the posterior distribution $p_{X|Y=y}$ using the stochastic denoising (also known as reverse) process of a diffusion model as a generative prior $p_X$ and a measurement likelihood function $\mathcal{L}(x|y) = p_{X=x|Y}(y)$, where $X$ is a generated image and $y = \mathcal{A}(x_0)$ is a known forward measurement. Assuming isotropic Gaussian noise on the measurement, the log likelihood is $\log \mathcal{L}(x|y) = Constant - \frac{1}{\sigma^2}\|y - \mathcal{A}(x)\|_2^2$. The discrete-time sampling process iterates from $t = T$ to 0, where the variable $x_t$ denotes the noised image and $\hat{x}_0(x_t) = \mathbb{E}[x_0|x_t]$ denotes the expected denoised version under the model. The DPS algorithm combines a denoise step $x'_{t-1} \leftarrow f_t(x_t, z_t)$, where $f_t$ incorporates the denoising model with the scheduled noising with $z_t$, with a gradient ascents step of the log-likelihood function

$$x_{t-1} \leftarrow x'_t - \zeta \nabla_{x_t}\|y - \mathcal{A}(\hat{x}_0(x_t))\|_2, \tag{19}$$

where $\zeta$ is a fixed step size that absorbs the forward measurement noise as well as normalizing the gradient of the squared norm by the value $\|y - \mathcal{A}(\hat{x}_0(x_t))\|_2$ as implemented by (Chung et al., 2022). We note that this corresponds to a Laplacian noise model $\log \mathcal{L}(x|y) = Constant - \zeta\|y - \mathcal{A}(x)\|_2$.

As part of the SLiCS framework, we propose a novel forward measurement model to sample from a concept-specific subspace. In this case, $Y \in \{1, \ldots, S\}$ denotes the concept of interest ($y = j$), and $\log \mathcal{L}(x|j) = Constant - \zeta\|\phi(x) - \mathcal{G}_j(\phi(x))\|_2$, where $\phi$ is the pre-trained embedding network (with $\ell_2$-normalization) and $\mathcal{G}_j$ is an implicit function that project's $\phi(x)$ to $\tilde{v}_j$ as in equation 3. Clearly, the log-likelihood function is maximized if the embedding lies with the positive cone such that it can be fully explained by $\mathcal{B}_j$ and decreases with increasing distance. By using this measurement model, we can sample images from the concept-specific cones to understand the SLiCS model.

$$x_{t-1} \leftarrow x'_t - \zeta \nabla_{x_t}\|\phi(\hat{x}_0(x_t)) - \mathcal{G}_j(\phi(\hat{x}_0(x_t)))\|_2, \tag{20}$$

Using the DINOv2 ViT-B/14 embeddings and the SLiCS model trained on MS COCO, we generated 5 images for each fo the 12 concepts, with $\zeta = 2$. The posterior samples are presented in Fig. 11. We include the codes of visualization based on the original implementation in the supplementary material.

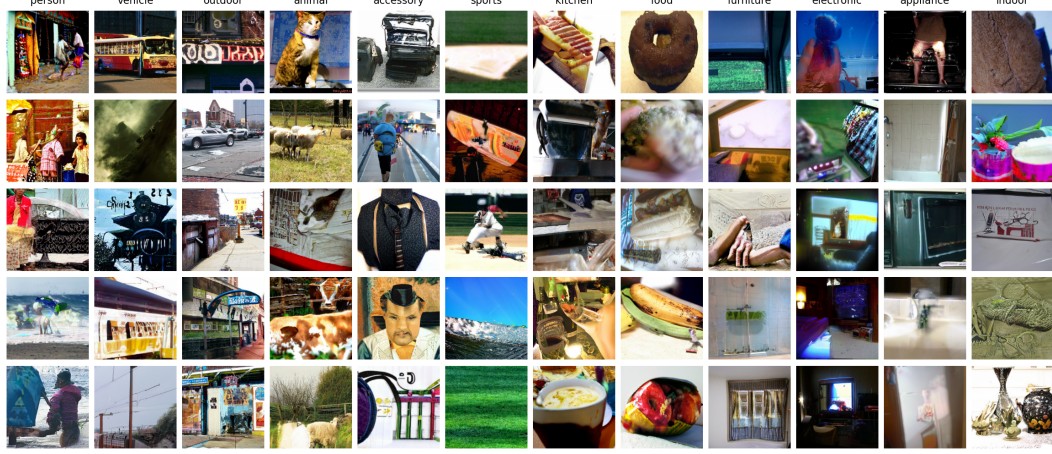

Figure 11: The DPS sampling results from the cone with DINOv2 ViT-B/14 embeddings (MS COCO). Each columns contains five samples of one concept.

Although this approach encourages posterior samples to come from the cone, it does not consider whether the coefficients are prototypical. A more nuanced approach would be to model the conditional distribution of coefficients given the concept using naive Bayes with a simple exponential distribution as done in prior work Grosse et al. (2012). The resulting likelihood would encourage the generation of images with coefficients for the atoms to match the model.

### A.2.6 CONCEPT DISTANCES AND DIVERGENCES

We compute the distance between the concept dictionaries and divergence between components for supervised SLiCS on the DINO embeddings of the COCO dataset using different numbers of atoms.

The Hausdorff distances between $\{\mathbf{B}_j\}_{j=1}^S$ are shown in Fig. 12. Together with Fig. 7, it shows that with an increasing dictionary size, the cone associated with different concepts tend to have more overlap, and the cosine similarity between the reconstructed and original embeddings increases with an increasing number of atoms. The results confirm the balance needed for the number of atoms: using too few prevents accurate reconstructions while too many blurs the boundaries between concepts.

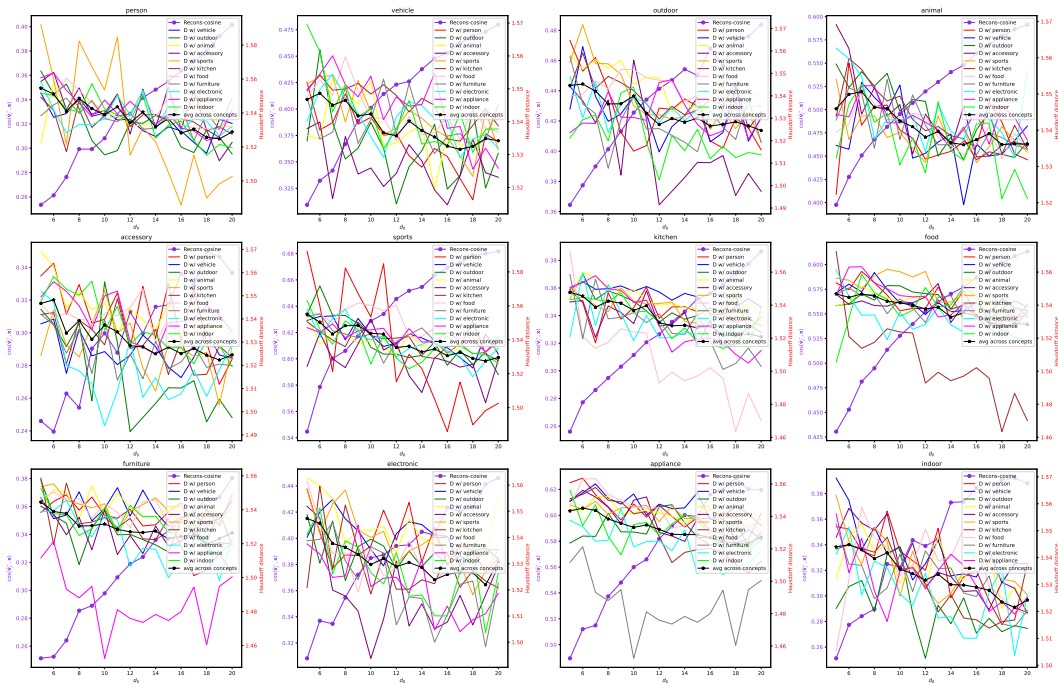

Figure 12: Hausdorff distances between concept-associated cones for different $d_0$, the number of atoms per concept, learned from DINOv2 embeddings on MS COCO. It is clear that when $d_0$ increases, in order to have more diversity, cones become closer to each other. While with more atoms the cosine similarity between $\tilde{\mathbf{v}}_j$ and $\mathbf{x}$ decreases.

Using examples from the training set, we present the Maximum Mean Discrepancy (MMD) (Gretton et al., 2012) between two samples of components from different concepts or different sub-label sets of the same concept. Fig. 13 displays distances between two samples of components from different concepts $j \neq j'$, i.e., $\{\mathbf{v}_{j,i}\}_{i \in \mathcal{I}_j}$ and $\{\mathbf{v}_{j',i}\}_{i \in \mathcal{I}_{j'}}$. Depending on the pairs of concepts, the divergence decreases with increasing $d_0$ or increases but then saturates. This indicates that fewer atoms better distinguishes some concepts, and more atoms deteriorates the ability to distinguish different concepts. Fig. 14 displays the divergence between samples of components from the same concept, but with different sub-labels. Here, the divergence increases (with marked steps) for increasing $d_0$, which indicates more atoms can help push sub-concepts further apart. Together these results indicate there is a balance to best separate components of different concepts and sub-concepts, which is confirmed in the validation performance as in Fig. 5.

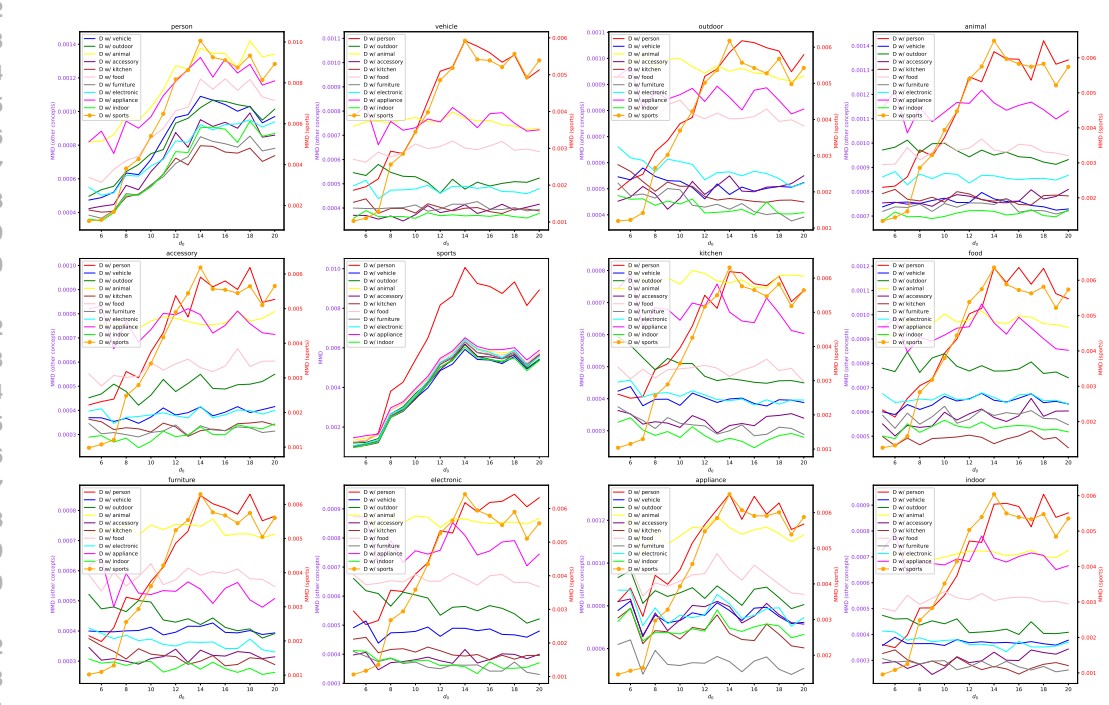

Figure 13: Maximum mean discrepancy between $\mathbf{v}_j$ of different concepts on a randomly sampled subset of the training set (DINO embedding on MS COCO). The sample size is 1000. The distances concerning the concept "sports" are on another scale, so it is plotted with a separate axis.

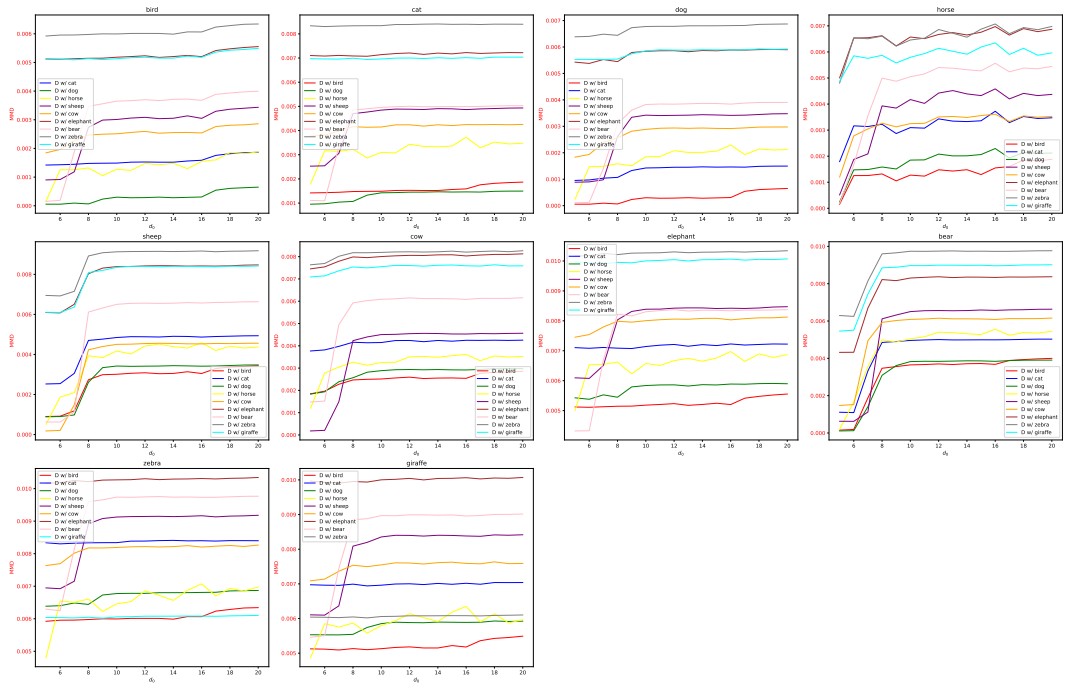

Figure 14: Maximum mean discrepancy between $\mathbf{v}_j$ of different sub-concepts within the concept "animal" on the training set (DINO embedding on MS COCO).

### A.2.7 CONCEPT COEFFICIENT STATISTICS

The distribution and correlation among coefficients for different concepts for supervised SLiCS on the DINO embeddings of the COCO dataset are shown in Fig. 15 and Fig. 16, respectively. Only the 'sports' concept has negative correlation between atoms, indicating mutually exclusive atoms. 'Indoor' and 'animal' are some concepts with apparent block structure.

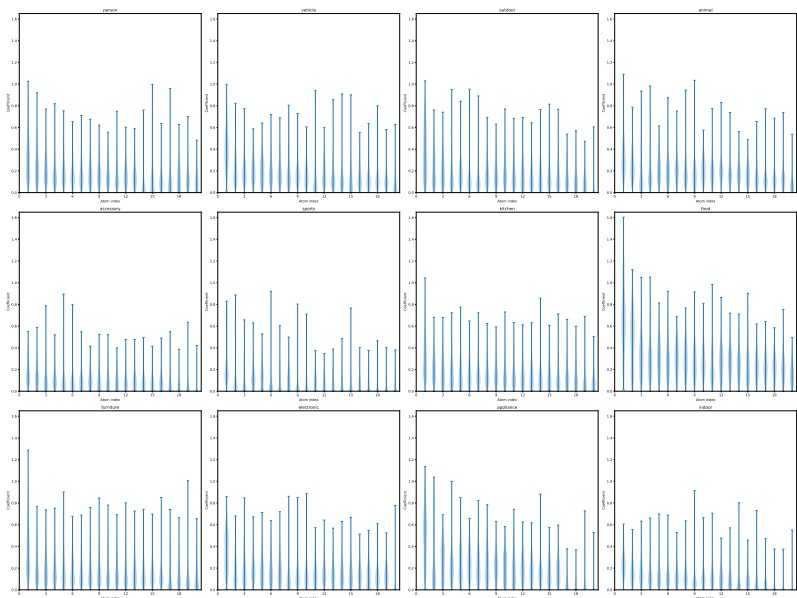

Figure 15: Distribution of the concept coefficients for each concept on the training set (DINO embedding on MS COCO) shown as violin plots in descending order of the mean.

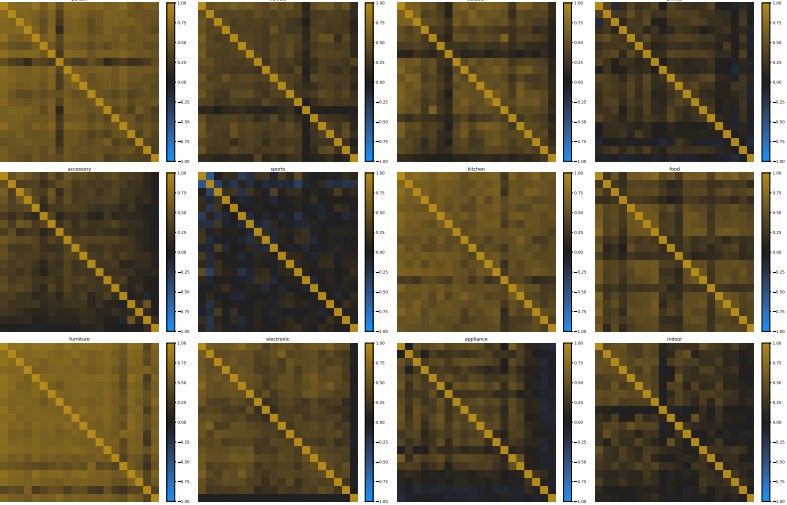

Figure 16: Correlation matrix of the concept coefficients for each concept on the training set (DINO embedding on MS COCO), calculated from the coefficient vector $\boldsymbol{\alpha}_j^i, i \in \mathcal{I}_j$ on samples with active concept $j$, in descending order of the mean.

### A.2.8 ADDITIONAL PERFORMANCE RESULTS

We present the results for an ablation study of non-negativity (removing the non-negative constraint on the coefficients $\alpha$) in Table 4. The results show that ablation performs worse and enforcing non-negativity is better for every case of S-SLiCS and U-SLiCS on the MS COCO dataset. However, ablating the non-negativity constraint for the supervised SVD approach improves the sub-label retrieval performance, but the general retrieval is still better for S-SVD with non-negativity.

Table 4: Performance results (mAP@20) for original (Orig.) and ablation of non-negativity (Abl.) on COCO dataset. **Gen.** denotes general label retrieval score; **Sub.** denotes sub-label retrieval score. Bold for best and underline for second best with ties for ±0.003.

| Model | | S-SVD | | S-SLiCS | | U-SLiCS | |
|---|---|---|---|---|---|---|---|
| | | Orig. | Abl. | Orig. | Abl. | Orig. | Abl. |
| CLIP ViT | Gen. | 0.905 > | 0.861 | **0.929** > | 0.910 | 0.871 > | 0.846 |
| | Sub. | 0.676 < | 0.707 | **0.742** > | 0.729 | 0.672 > | 0.652 |
| DINO | Gen. | 0.909 = | 0.909 | **0.984** ≈ | **0.987** | - | - |
| | Sub. | 0.728 < | 0.763 | **0.785** > | 0.774 | - | - |
| TiTok | Gen. | **0.823** > | 0.804 | 0.795 > | 0.740 | - | - |
| | Sub. | 0.421 < | **0.441** | 0.382 > | 0.331 | - | - |

We also explore alternative approaches for selecting the dimensionality of each concept $M_j$ and the number of active pseudo-labels for unsupervised (zero-shot) SLiCS with results in Table 6. Specifically, instead of giving the same $d_0$ for all concepts, we pick $M_j$, the number of atoms for concept $j$, during the SVD initialization such that the top $M_j$ singular values explain a given proportion of variance explained (PVE). (It is actually proportion of error explained as it is squared norm of the reconstruction error and the mean is not removed.) In an alternative method, $M_j$ is set to be proportional to the size of $\mathcal{I}_j$ for each concepts. For simplicity, the hyperparameters of the alternative methods are chosen such that total number of atoms $M$ is approximately the same as the original results, but another search grid search for the total number of atoms $M$ could be conducted. We present $M_j$ for these two cases in Table 5. The PVE method yields $M_j$ that is ±3 from the baseline on CLIP and from -8 to +5 on DINO. The PCF approach yields $M_j$ ranging from 5 to 49 on CLIP and from 6 to 57 on DINO.

Table 5: The number of atoms for each concept $M_j, j \in \{1, \ldots, S\}$, when chosen to be proportional to variance explained (PVE) or to concept frequency (PCF) for CLIP ViT-B/32 and DINO on MS COCO, compared to constant $M_j = d_0$ of 17 for CLIP and 20 for DINO. It should be noted that in practice, rounding was used without allocating reminders. This means such that the total number of atoms is not exactly equal to the original $S \cdot d_0 = 12d_0$: 204 for CLIP and 240 for DINO.

| Model | | $M_1$ | $M_2$ | $M_3$ | $M_4$ | $M_5$ | $M_6$ | $M_7$ | $M_8$ | $M_9$ | $M_{10}$ | $M_{11}$ | $M_{12}$ | Sum |
|---|---|---|---|---|---|---|---|---|---|---|---|---|---|---|
| ViT | PVE | 20 | 16 | 19 | 17 | 20 | 17 | 14 | 16 | 15 | 20 | 17 | 16 | 207 |
| | PCF | 49 | 19 | 9 | 17 | 13 | 17 | 15 | 12 | 22 | 9 | 5 | 11 | 198 |
| DINO | PVE | 24 | 22 | 19 | 19 | 25 | 10 | 22 | 16 | 23 | 21 | 12 | 25 | 238 |
| | PCF | 57 | 23 | 11 | 20 | 15 | 20 | 18 | 14 | 26 | 11 | 6 | 13 | 234 |

As shown in Table 6, using PVE does not meaningfully alter performance; only on the U-SLiCS case was PVE better than than original by more the 0.003. The PCF approach improves the S-SVD baseline, but is approximately the same for S-SLiCS and slightly worse for U-SLiCS. This also shows the robustness of SLiCS to the distribution of atoms across concepts.

For unsupervised SLiCS, we also try dynamic pseudo-labeling where a threshold on probabilities created by passing the cosine similarities of the text embeddings to a softmax function creates a different number of active concepts per instance. The accuracy for a different number of active concepts is shown in Fig. 17. We pick the threshold for the zero-shot classification scores to infer the presence of a concept such that the inferred average number of concepts on the validation set is equal to $\bar{S} = 2.303$. The dynamic allocation is slightly better at the general-label level, and slightly worse for the sub-label level.

More detailed quantitative results and qualitative results are shown in Table 7, Fig. 18 and Fig. 19.

Table 6: Performance results (mAP@20) on COCO dataset comparing the following: original (**Orig.**) results with fixed $d_0$ and fixed number of active pseudo-labels for unsupervised SLiCS (U-SLiCS); **PVE** when the number of atoms $M_j$ is chosen to explain a fixed proportion of variance explained in the initialization; **PCF** when $M_j$ is proportional to concept frequency; and **Dyn.** when a different number of active pseudo-labels are chosen for each instance in U-SLiCS. **Gen.** denotes general label retrieval score; **Sub.** denotes sub-label retrieval score. Bold for best and underline for second best with ties for ±0.003.

| Model | | S-SVD | | | S-SLiCS | | | U-SLiCS | | | |
|---|---|---|---|---|---|---|---|---|---|---|---|
| | | Orig. | PVE | PCF | Orig. | PVE | PCF | Orig. | PVE | PCF | Dyn. |
| ViT | Gen. | $0.905 \approx\approx$ | 0.903 | 0.906 | $\underline{\textbf{0.929}} \approx\approx$ | $\underline{\textbf{0.927}}$ | $\underline{\textbf{0.929}}$ | $0.887 >>>$ | 0.885 | 0.866 | 0.881 |
| | Sub. | $0.676 \approx<$ | 0.677 | 0.683 | $\underline{\textbf{0.742}} \approx\approx$ | $\underline{\textbf{0.743}}$ | $\underline{\textbf{0.743}}$ | $0.673 \approx>>$ | 0.670 | 0.657 | 0.666 |
| DINO | Gen. | $0.909 \approx<$ | 0.906 | 0.912 | $\underline{\textbf{0.984}} \approx\approx$ | $\underline{\textbf{0.983}}$ | $\underline{\textbf{0.986}}$ | - | - | | - |
| | Sub. | $0.728 \approx<$ | 0.726 | 0.732 | $\underline{\textbf{0.785}} >\approx$ | 0.778 | $\underline{\textbf{0.783}}$ | - | - | | - |

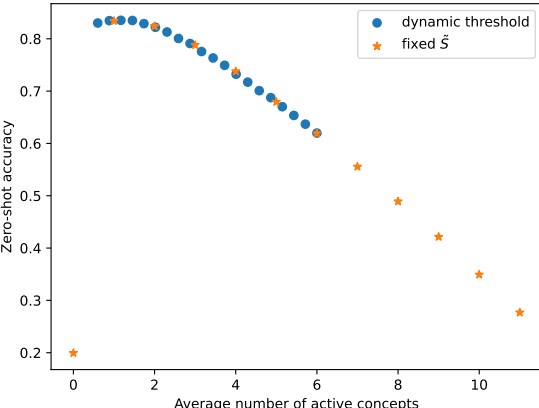

Figure 17: CLIP ViT-B/32 zero-shot accuracy versus average number of active concepts on validation set of COCO, calculated with different thresholds for dynamic or fixed integers for $\tilde{S}$.

Table 7: Quantitative results (mAP@20) of each concept of MS COCO for UF-D (unfiltered retrieval with DINO embeddings), S-SLiCS-D (supervised SLiCS with DINO embeddings), UF-CLIP (unfiltered retrieval with CLIP ViT-B/32 embeddings) and S-SLiCS-C (supervised SLiCS with CLIP ViT-B/32 embeddings).

| Name | UF-D | S-SLiCS-D | UF-CLIP | S-SLiCS-C |
|---|---|---|---|---|
| person | 0.859 | 0.995 | 0.884 | 0.987 |
| vehicle | 0.826 | 0.998 | 0.819 | 0.961 |
| vehicle (sub) | 0.776 | 0.834 | 0.759 | 0.831 |
| outdoor | 0.705 | 0.946 | 0.643 | 0.842 |
| outdoor (sub) | 0.615 | 0.723 | 0.525 | 0.650 |
| animal | 0.921 | 0.999 | 0.887 | 0.983 |
| animal (sub) | 0.905 | 0.903 | 0.833 | 0.859 |
| accessory | 0.583 | 0.949 | 0.550 | 0.822 |
| accessory (sub) | 0.458 | 0.596 | 0.455 | 0.595 |
| furniture | 0.704 | 0.990 | 0.710 | 0.868 |
| furniture (sub) | 0.645 | 0.666 | 0.630 | 0.686 |
| electronic | 0.722 | 0.994 | 0.710 | 0.900 |
| electronic (sub) | 0.642 | 0.787 | 0.610 | 0.712 |
| appliance | 0.780 | 0.963 | 0.753 | 0.948 |
| appliance (sub) | 0.722 | 0.892 | 0.672 | 0.829 |
| indoor | 0.660 | 0.971 | 0.692 | 0.890 |
| indoor (sub) | 0.582 | 0.776 | 0.619 | 0.700 |
| Avg. | 0.793 | 0.985 | 0.791 | 0.931 |
| Avg. (sub) | 0.698 | 0.785 | 0.671 | 0.744 |

27

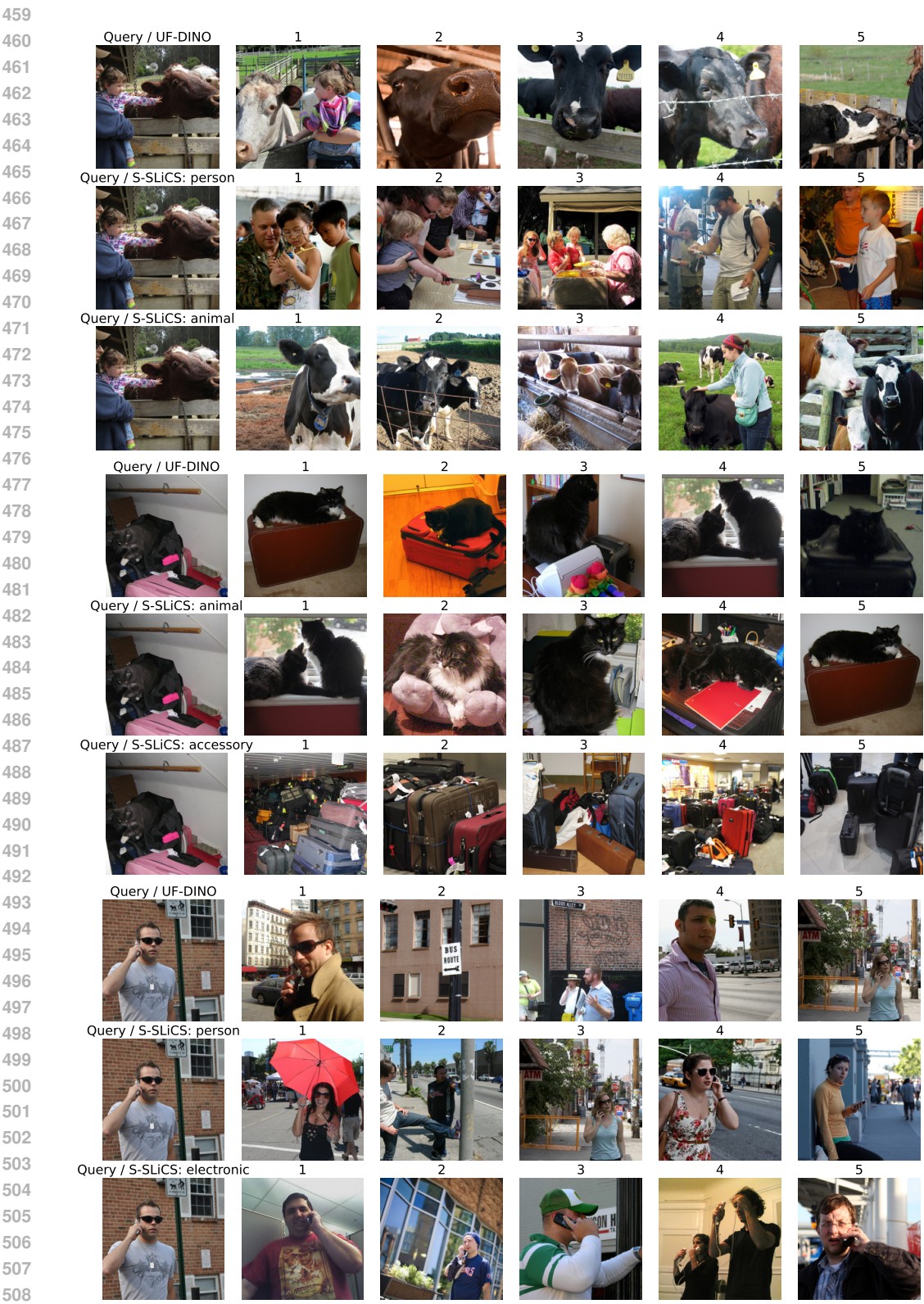

Figure 18: Top-5 retrieved images on MS COCO for DINO embeddings.

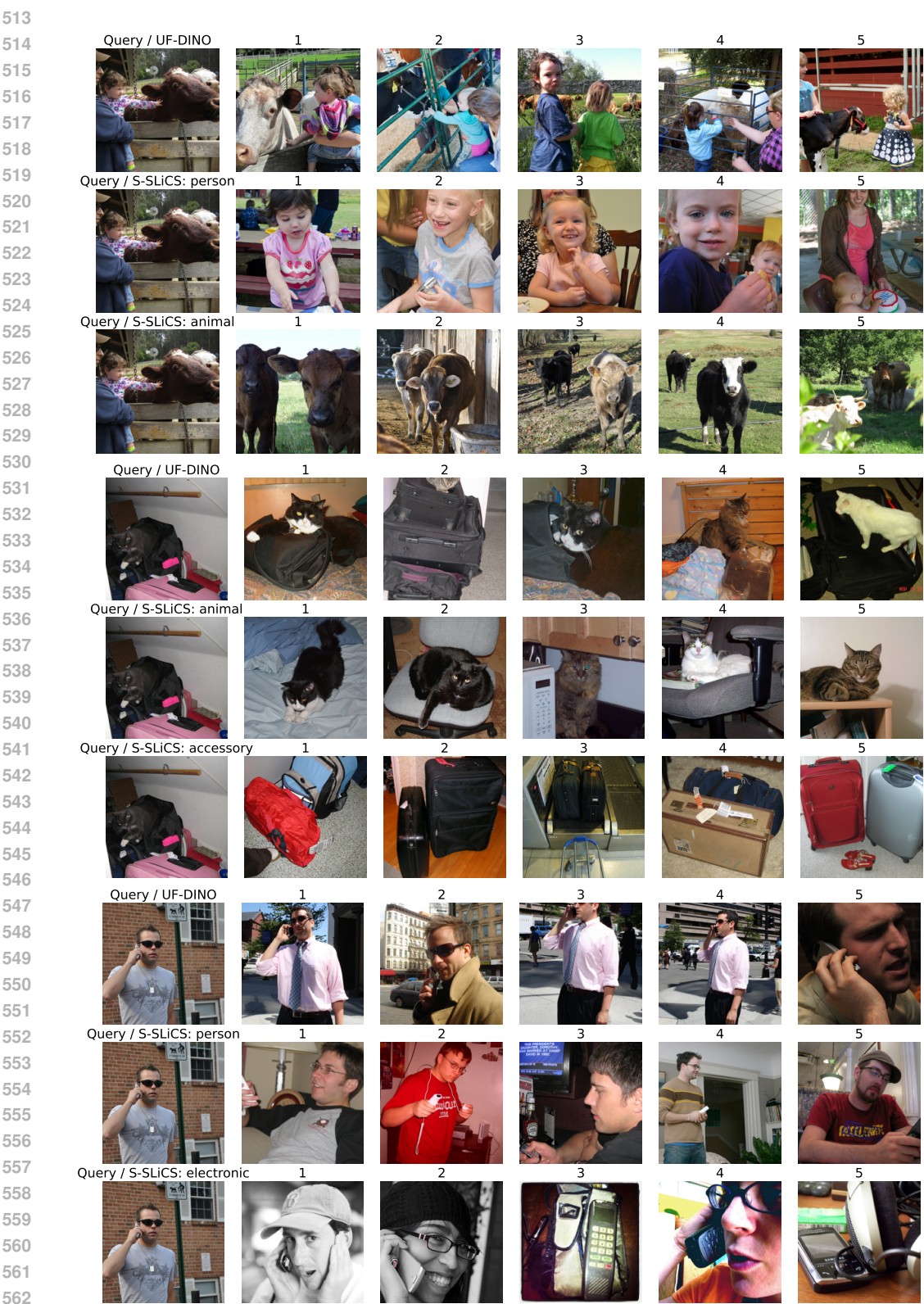

Figure 19: Top-5 retrieved images on MS COCO for CLIP ViT-B/32 embeddings.

## A.3 INTERPRETATION OF ATOMS

In order to interpret the learned dictionary, or the atoms specifically, along the lines of previous work (Zhang et al., 2021; Fel et al., 2023), we use atoms as queries to retrieve patches constructed from images with active concepts associated with the atoms. We pick 1,000 images with active concept $j$ from the MS COCO training set, extract five 64 by 64 patches from each image. The atoms from $M_j$ are then treated as queries to retrieve the most similar patches. In this way, we can show the visual aspects from each concept that each atom focuses on. We present the top-5 retrieval results for the first 5 atoms from each $M_j$ in Fig. 20 and Fig. 21. Fig. 20 shows the results under the score $\cos(\mathbf{b}_i, \mathbf{z}_i)$, where $\mathbf{b}_i$ is the atom and $\mathbf{z}_i \in \mathbb{R}^{64 \times 64}$ is the patch. Fig. 21 shows the results under the score $\boldsymbol{\alpha}$ when reconstructing $\tilde{\mathbf{v}}_j$.

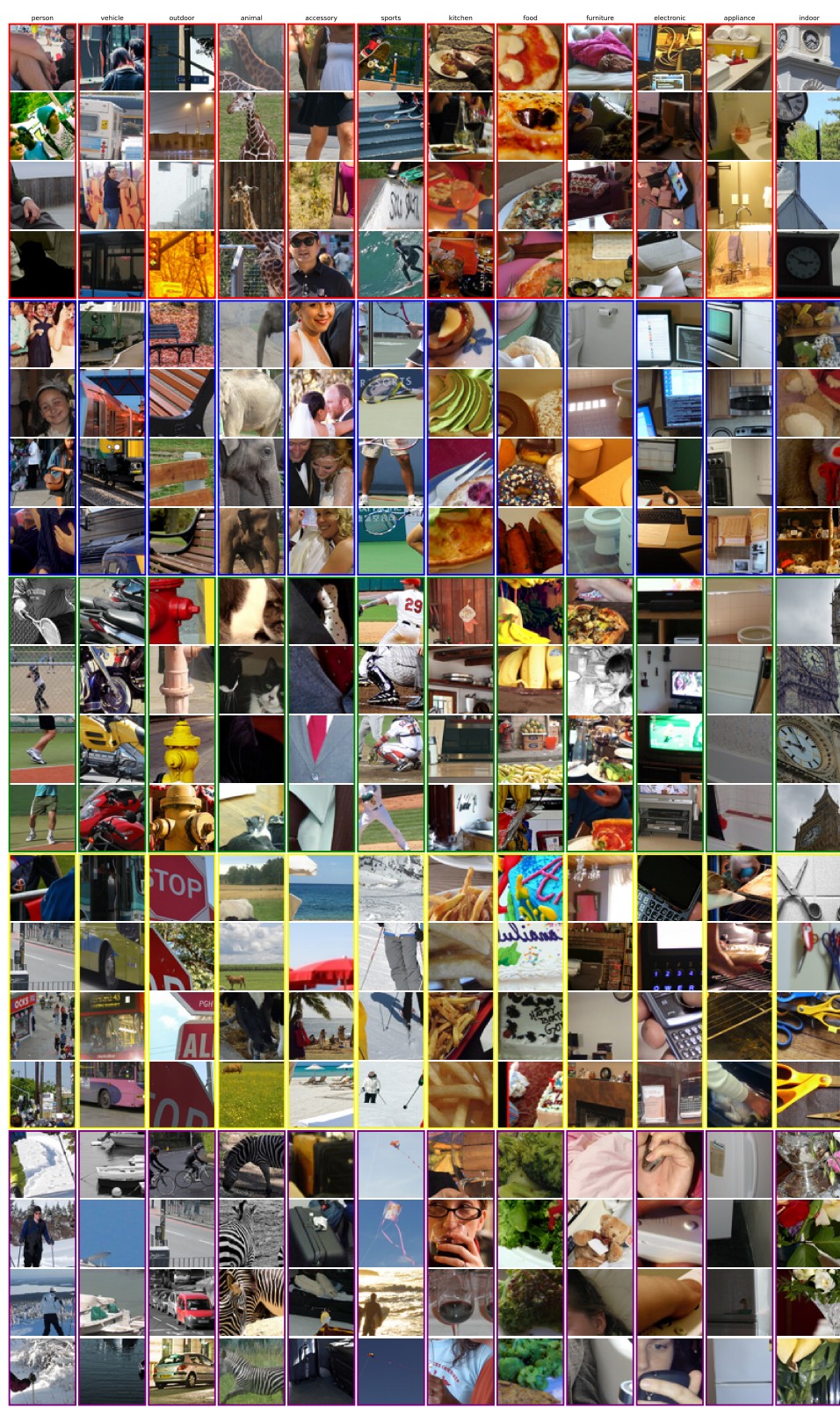

Figure 20: Interpretability of the atoms. Each column shows the retrieval results for five atoms from a concept. Top-5 retrieved patches are presented for each atom, in a block outlined with different colors. The retrieval score is the cosine similarity between the query and the patches.

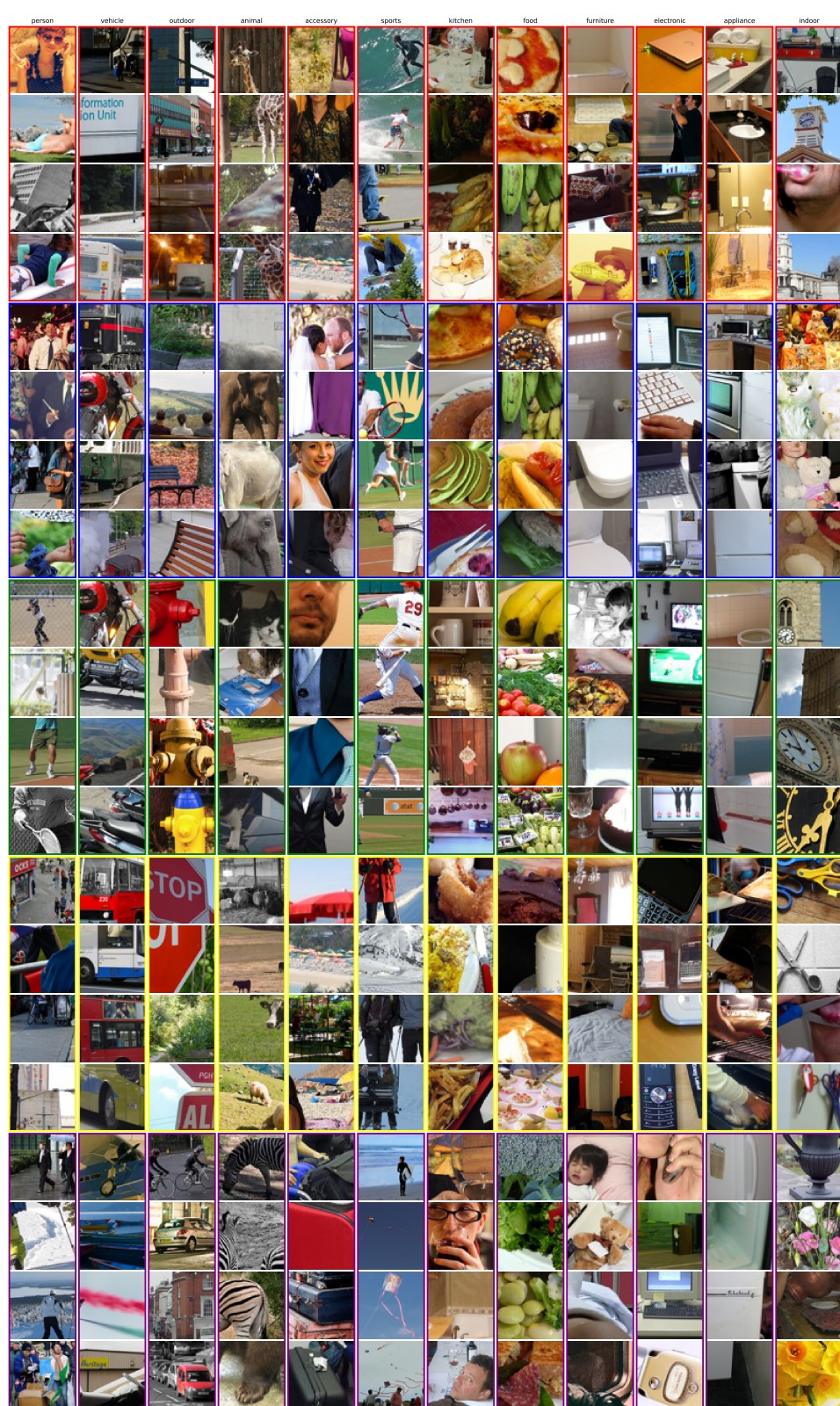

Figure 21: Interpretability of the atoms. Each column shows the retrieval results for five atoms from a concept. Top-5 retrieved patches are presented for each atom, in a block outlined with different colors. The retrieval score is the coefficient corresponding to each atom when reconstructing $\tilde{\mathbf{v}}_j$.

