# OpenReview forum: "Disentangling Latent Embeddings with Sparse Linear Concept Subspaces (SLiCS)"
_ICLR.cc/2026/Conference — Submitted to ICLR 2026_

### Official Review · Reviewer_V1fX · 2025-10-24

**Soundness:** 3
**Presentation:** 3
**Contribution:** 2
**Rating:** 4
**Confidence:** 4

**Summary:**

The paper presents a novel approach to refining global embeddings from image embedding models like CLIP into more structured, two-tiered concept-labeled subspaces, with a focus on improving "concept filtered retrieval." The authors employ dictionary learning to develop a set of central high-level concepts, each consisting of atoms that represent sub-concepts. This approach transforms the embedding space into a collection of distinct subspaces or "cones," each containing relevant sub-concepts. The methodology is termed SLiCS - Sparse Linear Concept Subspaces.


The technique involves approximating each embedding vector as a linear combination of concepts from various subspaces, with atom-level representations organized within each concept subspace. A non-negative constraint is applied to the coefficient vector, forming a "cone" that encourages the co-occurrence of positively similar concepts, aligning with the cosine similarity metric utilized in CLIP models.


In its supervised variant, the approach minimizes the mean squared error of the reconstructed embedding approximation. This is achieved under a non-negative constraint for co-efficients using a modified supervised dictionary learning algorithm based on K-SVD, employing a rank-1 approximate SVD-based update heuristic. An unsupervised version involves inferring pseudo label mapping from a pre-existing concept list, derived from CLIP embeddings, followed by the same supervised process.


The authors evaluate their method using training and evaluation sets from MIRFlickr25K and MS COCO, comparing retrieval performance against CLIP, DINO, TikTok, and SpLiCE, with mean average precision at top 20 as the metric. The findings demonstrate a significant improvement of SLiCS models over other methods across all evaluated embeddings.

**Strengths:**

- The methodology employs advanced dictionary learning techniques to map flat embeddings to structured concept spaces.
- The algorithm is thoroughly detailed in Appendix 1, offering clear and comprehensive explanations.
- The paper provides a robust mathematical analysis of the learning method, including alternatives and theoretical guarantees.
- It includes illustrative visualizations, such as image-to-prompt examples for various methods.
- Progressive visualizations of intermediate data points within a subspace are provided, offering qualitative insights into the learned concepts and atoms.
- The study includes a comparative analysis against SpLiCE, one of the most relevant and comparable methods.

**Weaknesses:**

The problem formulation of "concept filtered retrieval" seems somewhat contrived. The purpose of embedding models like CLIP is to encapsulate the comprehensive semantics of an image. Consequently, if the query is with an image of a "man talking on phone," the retrieval task focusing separately on images of men and phones may not be practically relevant. While the authors have developed an intricate mapping scheme between a flat CLIP embedding and a structured concept space, the practical relevance of this mapping for retrieval tasks remains unclear to me.

**Questions:**

- Curious, if you have any aggregate analysis on how distinct/redundant the atoms within a subspace is?
- Also if there was any attempt or analysis done to auto-label the fine grained concept components, as in some of the explainable ML efforts.

---

> ### Author Response · Authors · 2025-11-25
>
> We thank the reviewer for the thoughtful summary, noted strengths, and constructive criticism.
>
> **Weakness**
> > The problem formulation of "concept filtered retrieval" seems somewhat contrived. [...] if the query is with an image of a "man talking on phone," the retrieval task focusing separately on images of men and phones may not be practically relevant. [...] the practical relevance of this mapping for retrieval tasks remains unclear to me.
>
> SLiCS is motivated by the image retrieval of large image datasets consisting of complex scenes. When scenes are very complex, it is easier to retrieve images that have high partial similarity to the query (i.e., some concepts) than to retrieve an image that is similar comprehensively, especially when one cares about the specific concept in the query than the comprehensive semantics of the image. Such concept-filtered tasks are not contrived, but rather widespread for such one-shot tasks.   SLiCS provides an intuitive method for doing this.
>
> Another application is to cluster the components of each concept to understand variation of a concept ignoring the rest of the scene, analogously to how image segmentation is used by Ghorbani et al. (2019) to discover concepts associated with a class. Without disentanglement or input space segmentation, clusters of images with a concept without disentanglement may be less informative due to the other active concepts.
>
> We have added a discussion with these applications.
>
>
> **Q1.**
> >  if you have any aggregate analysis on how distinct/redundant the atoms within a subspace is?
>
> In Figure 7 in the Appendix we had reported that the cosine similarity of atoms within the same group was (−0.056±0.062) and that the cross-group similarity (0.000 ± 0.134). We have now confirmed that there is no redundancy of atoms for the same concept such that none are interior to the convex hull for all the tested different values of $d_0$. This was confirmed by showing that there is a non-zero residual when an atom is attempted to be reconstructed from the remaining atoms.
>
> **Q2.**
> > Also if there was any attempt or analysis done to auto-label the fine grained concept components, as in some of the explainable ML efforts.
>
>  That is a great suggestion. One approach would be to cluster the components of a concept (as mentioned above) and then find the text embeddings that are best fit by this cluster. This clustering approach would provide a more fine-grained interpretation of the data manifold. Of course, the selection of an appropriate number of clusters may be difficult, but model selection criterion could be used.

---

### Official Review · Reviewer_KU5w · 2025-10-31

**Soundness:** 3
**Presentation:** 3
**Contribution:** 2
**Rating:** 4
**Confidence:** 3

**Summary:**

The paper introduces SLiCS, a method that targets concept-filtered image retrieval. It uses a K-SVD-like alternating optimization to learn a dictionary of concepts, where each concept is composed of atoms. This dictionary is used to linearly decompose image embeddings. The approach is evaluated on MIRFlickr-25K and MS-COCO using CLIP (transformer and CNN-based), DINOv2, and TiTok embeddings, reporting gains in mAP@20 with advantage over baselines.

**Strengths:**

1) Clear motivation. Concept-filtered retrieval is well-posed and seems to be practically useful.
2) Intuitive and clearly presented method. The approach is straightforward and well-explained; the optimization scheme and underlying intuition are easy to follow.
3) Broad evaluation across embedding types. The method is tested on several backbones (CLIP, DINOv2, TiTok) and datasets, suggesting generality across embedding spaces.
4) Inherent interpretability. The per-concept subspaces lend themselves naturally to visualization and qualitative inspection, which enhances transparency of retrieval results.

**Weaknesses:**

1) Limited novelty. The proposed method is essentially a variant of K-SVD with non-negativity and concept-wise group sparsity. While it is nicely adapted to the retrieval context, the conceptual advance over classical dictionary learning or prior works like SpLiCE is relatively small.
2) Restricted evaluation. The experiments are limited to two datasets and a single retrieval metric (mAP@20). Including additional metrics such as Recall@K or evaluating on larger/more diverse datasets would strengthen the empirical evidence.
3) Potential hyperparameter sensitivity. The number of atoms per concept and the number of active concepts per image appear to be tuned on validation sets, but there is little analysis of how performance varies with these choices. The approach may be sensitive to these hyperparameters.

**Questions:**

1) You mention setting the number of active concepts to 2, motivated by the average number of concepts per image (2.467 and 2.303). Why not choose a larger value and allow the model to suppress irrelevant concepts through sparsity? Would the method naturally drive extra concept coefficients to zero?
2) Is there a systematic or data-driven way to select the number of concepts or atoms per concept based solely on the embeddings (e.g., reconstruction error trends or stability analysis)?
3) How does performance change if these hyperparameters are misspecified?

---

> ### Author Response · Authors · 2025-11-25
>
> **Weakness 1.**
> > conceptual advance over classical dictionary learning or prior works like SpLiCE is relatively small
>
> The use of concept labels (supervised or zero-shot) that informs the group sparsity is a unique aspect of our algorithm that is completely distinct from both existing dictionary learning algorithms and from SpLiCE. Given the concept labels, the mechanics of the update are similar to K-SVD, but the assumptions of existing dictionary learning or SpLiCE are individual sparsity, in comparison we use group activations to learn meaningful subspaces that co-occur. Thus, while unsupervised dictionary learning with block sparsity exists, we are not aware of any prior work on algorithms for supervised block sparse learning. The fact that concepts co-occur also makes our algorithm distinct from subspace clustering which assumes each instance belongs to one and only one subspace. The atoms in defining concept subspaces compete during co-active concepts to better disentangle the components.   Combined with the ablation study, it shows that SLiCS use of non-negative and modeling co-active concepts (compared to S-SVD baseline) provides meaningful gains in concept-filtered retrieval.
>
> **Weakness 2.**
> > The experiments are limited to two datasets and a single retrieval metric (mAP@20). Including additional metrics such as Recall@K or evaluating on larger/more diverse datasets would strengthen the empirical evidence.
>
> We note that recall@K for small K is proportional to precision@K when the number of relevant instances is larger than K. The average precision at K (AP@K) metric is more stringent than precision at K (and thus recall@K).  We can certainly compute more performance metrics, but we expect the results to be concordant. We agree that more datasets would always strengthen the case, and are planning on identifying datasets that have more labeled concepts or use zero-shot labeling as pseudo-ground truth to evaluate concept disentanglement for our own user-defined concepts. However, we believe the existing results and novelty present clear evidence of the utility of concept-specific retrieval and generation.
>
> **Weakness 3.**
> >  The number of atoms per concept and the number of active concepts per image appear to be tuned on validation sets, but there is little analysis of how performance varies with these choices. The approach may be sensitive to these hyperparameters.
>
> The validation curves for number of atoms per concept were provided in the appendix, see Figure 5. The performance only varies by 0.01 around the selected dimensionality, which is to say it is not very sensitive. We now report the performance when the number of active concepts for unsupervised SLiCS varies in Figure 17.
>
> **Q1.**
> >  Why not choose a larger value and allow the model to suppress irrelevant concepts through sparsity? Would the method naturally drive extra concept coefficients to zero?
>
> For the unsupervised zero-shot case, we now calculate the accuracy for different numbers and different thresholds in Figure 17. Note that our dictionary learning algorithm does not explicitly encourage further sparsity (beyond what is induced by the non-negative constraint). However, if the zero-shot generated a false positive then there is something about the embedding that indicates that concept. It is too optimistic to hope that the dictionary wouldn’t try to learn from this false association. It is possible to use a higher number of active pseudo-labels and then have an additional hyper-parameter on a group LASSO for concept coefficient vectors. It is really a question whether the zero-shot is going to be better than the generative classifier formed by saying it is active if the concept can be well explained. After training, inference of active concepts through group LASSO may be competitive with the zero-shot, but during learning it is possible that the active concept inference is going to be noisier than the zero-shot.
>
> **Q2.**
> > Is there a systematic or data-driven way to select the number of concepts or atoms per concept based solely on the embeddings (e.g., reconstruction error trends or stability analysis)?
>
> The numbers of atoms per concept can be chosen in the SVD-based initialization from choosing the rank such that the criterion covers a proportion of the variance (or as suggested proportional to the frequency of the concept). We report results for both of these and note that the results do not change much.
>
>
> **Q3.**
> > How does performance change if these hyperparameters are misspecified?
>
> As noted for Weakness 3, the validation curves were provided in the appendix, see Figure 5. The performance only varies by 0.01 around the selected dimensionality, which means it is not too sensitive near the optimal.

---

> ### Author Response · Authors · 2025-12-03
>
> **Q4.**
> >Restricted evaluation. The experiments are limited to two datasets and a single retrieval metric (mAP@20).
>
> We report in Tab. 7 the quantitative results for each concept for unfiltered retrieval and two types of S-SLiCS. It is a more detailed look into the concept-specific performances. Some concepts, like 'electronic', are difficult for SLiCS to separate apart the sub-concepts, hence slightly lower mAP scores in sub-label retrieval across all models. Other concepts, like 'person', are more general and common for the pre-trained network to extract the concept-specific features during the training process. Therefore, it obtains a higher score.

---

### Official Review · Reviewer_jQMf · 2025-11-01

**Soundness:** 2
**Presentation:** 3
**Contribution:** 2
**Rating:** 4
**Confidence:** 4

**Summary:**

The paper “Disentangling Latent Embeddings with Sparse Linear Concept Subspaces (SLiCS)” introduces a novel approach for decomposing dense vision-language embeddings (like CLIP’s) into interpretable, concept-specific subspaces. The authors propose a supervised dictionary learning model in which each image embedding is represented as a non-negative, sparse combination of basis vectors (“atoms”) grouped by semantic concepts. This decomposition allows each component of an embedding to capture a specific concept, effectively disentangling complex scene representations into meaningful substructures. The optimization framework extends K-SVD with non-negativity and group sparsity constraints, ensuring convergence while maintaining interpretability. In addition, an unsupervised variant leverages CLIP’s text embeddings to infer pseudo-labels via zero-shot classification, enabling SLiCS to operate even without annotated data.

**Strengths:**

1. SLiCS introduces a principled method to decompose dense, opaque vision-language embeddings into interpretable concept-specific subspaces. Unlike previous methods that rely directly on word embeddings or linear projections, SLiCS learns structured, non-negative, sparse subspaces that correspond to semantically coherent concepts. This improves both interpretability and controllability of latent representations — a major step forward for understanding how high-dimensional embeddings encode different aspects of scenes.
2. The paper presents a well-founded optimization framework inspired by K-SVD but extended to enforce non-negativity and group sparsity, ensuring each subspace captures a distinct concept. The authors also provide guaranteed convergence for their alternating optimization procedure and discuss computational considerations like mini-batch training for scalability. This level of mathematical rigor and algorithmic detail strengthens the credibility of the proposed method.
3. A strong point of SLiCS is its broad applicability. It can operate on embeddings from different models (CLIP, TiTok, DINOv2) and handle both supervised and unsupervised settings. The unsupervised variant cleverly leverages CLIP’s text-aligned embeddings for zero-shot pseudo-labeling, making it practical for large datasets where concept labels may not be available.
4. SLiCS provides tangible improvements in concept-filtered image retrieval, a task where existing holistic embeddings often fail to separate distinct visual elements. The results demonstrate consistent gains in precision across datasets and embedding types, showing that disentangled embeddings are not just interpretable but also more effective for fine-grained retrieval and potentially for conditional generation.
5. By providing a method to explicitly identify and manipulate concept-level components within a latent space, SLiCS advances the broader goal of explainable and modular machine learning. It bridges the gap between powerful but black-box embedding models and transparent, concept-driven reasoning, offering new possibilities for interactive or human-in-the-loop AI systems.

**Weaknesses:**

1. While the paper provides a theoretically sound optimization framework, the dictionary learning and non-negative sparse coding steps can be computationally demanding, especially for high-dimensional embeddings or large-scale datasets. Although mini-batch training is mentioned as a workaround, the algorithm may still struggle to scale to modern foundation model embeddings (e.g., CLIP-L/14 or EVA-CLIP) or real-time applications.
2. Both the supervised and unsupervised versions of SLiCS rely on a fixed set of concept labels. In practice, defining this concept set can be subjective and may limit generalization to unseen or abstract concepts. The unsupervised variant partially addresses this by using zero-shot text embeddings, but it still assumes a known vocabulary and a fixed number of active concepts per image, which can constrain flexibility in open-world scenarios.
3. Although SLiCS aims to learn interpretable subspaces, the semantic meaning of each subspace is inferred post-hoc by matching atoms to text embeddings. This makes interpretability somewhat indirect and dependent on the quality and coverage of the text encoder, rather than emerging naturally from the model. It may also lead to mismatches between the learned subspace and human-understandable concepts.
4. The experiments focus primarily on concept-filtered image retrieval tasks, without broader exploration of other applications such as conditional generation, editing, or reasoning. While retrieval is a clean benchmark, it does not fully demonstrate whether SLiCS provides richer disentanglement properties or generalizes to more complex downstream uses.
5. The paper lacks comparison against modern interpretable embedding methods or representation disentanglement frameworks (e.g., Concept Bottleneck Models, Concept Whitening, or interpretable subspace discovery via CAVs). Including such baselines would better situate SLiCS within the broader landscape of interpretability research.
6. The algorithm’s performance depends on choices such as the number of atoms per concept (d₀) and initialization via truncated SVD. While the paper provides reasonable heuristics, it does not explore how sensitive SLiCS is to these design choices. This raises questions about robustness and reproducibility across datasets and embedding spaces.

**Questions:**

1. How sensitive is SLiCS to the choice of hyperparameters such as the number of atoms per concept (d_0) or the number of training iterations? Did you observe significant performance variance across different initializations?
2. The paper claims guaranteed convergence for the alternating optimization procedure. Could the authors elaborate on the nature of this guarantee — is it convergence to a stationary point, or to a global minimum under certain assumptions?
3. Given that modern embeddings (e.g., CLIP-L/14 or OpenCLIP-H) can have tens of thousands of dimensions, how does the computational cost of SLiCS scale with embedding size? Are there approximations or dimensionality reduction techniques that make it more practical?
4. The method enforces non-negative coefficients based on the intuition that cosine similarity defines semantic inclusion. Have the authors empirically compared this with unconstrained or signed decompositions to validate the impact of this constraint on interpretability and retrieval performance?
5. The paper assumes all concept dictionaries have the same number of atoms (M_j = d_0). Did the authors explore adaptive or data-driven ways to set M_j depending on concept complexity or label frequency?
6. How consistent are the learned subspaces across different runs or datasets? For instance, do the same concepts yield similar atom groups, or is there significant drift depending on initialization and data distribution?
7. In the unsupervised variant, pseudo-labels are derived from CLIP text embeddings. How reliable are these labels for ambiguous or fine-grained concepts, and do noisy pseudo-labels degrade subspace interpretability?
8. The model assumes additive decomposition across concepts. Have the authors considered whether certain concepts (e.g., “dog” and “grass”) interact non-linearly in the latent space, and if so, how might SLiCS capture such dependencies?
9. The paper demonstrates improved performance on concept-filtered retrieval. Have the authors tested SLiCS for other tasks — such as concept-based editing, few-shot learning, or conditional generation — to assess its general utility?

---

> ### Author Response · Authors · 2025-11-25
> **Weaknesses**
>
> We want to thank the reviewer for the detailed list of strengths, weaknesses. We apologize for the delay in our rebuttal, but we wanted to provide a comprehensive revision.
>
> **1**.
> > Although mini-batch training is mentioned as a workaround, the algorithm may still struggle to scale to modern foundation model embeddings (e.g., CLIP-L/14 or EVA-CLIP) or real-time applications.
>
> For inference, non-negative least squares algorithm typically scales linearly in the dimension $d$ and quadratically in the number of atoms $d_0$. We use SciPy’s standard non-negative least squares solver, which takes on average less than 4 ms for a 1024-dimensional vector with $d_0=60$ atoms. For training, rank-1 truncated SVD scales linearly with sample size $N$ and dimension $d$. The initial SVD of each concept vector is quadratic in the dimension $d$, but this is a one time cost. Going to 768 or 1024 dimensions associated to CLIP-L/14 and EVA-CLIP is not going to slow down as the ResNet-50 is already 1024.
>
> **2.**
> > this concept set can be subjective and may limit generalization to unseen or abstract concepts. The unsupervised variant partially addresses this by using zero-shot text embeddings, but it still assumes a known vocabulary and a fixed number of active concepts per image
>
>  Our results are limited to a known set of concepts, but it is simple to use a dynamic allocation of active concepts in the zero-shot pseudo-label by using a threshold, and to create variable-size dictionaries explaining a fixed proportion of variance based on the SVD of instances associated to each concept. We now report results with variable sizes dictionaries in the appendix, which do not show much difference.  Discovery of concepts could use completely unsupervised dictionary learning (Szabó et al. 2011) (cited in the revision) or other methods like in Ghorbani et al. (2019), which we now mention in Section 2.
>
> **3.**
> > the semantic meaning of each subspace is inferred post-hoc by matching atoms to text embeddings. This makes interpretability somewhat indirect and dependent on the quality and coverage of the text encoder, rather than emerging naturally from the model. It may also lead to mismatches between the learned subspace and human-understandable concepts
>
> The verbal interpretation of the learned concept subspaces using CLIP embeddings is based on finding text embeddings with the best reconstruction using the subspaces (the interpretation of individual atoms has not been investigated). Note that the embedding of longer sentences and blocks of text could be used to give finer grained interpretability. Additionally, semantic interpretation is also possible from the images generated by sampling from the cones via the Diffusion Posterior Sampling (DPS) as shown in Figure 11 in the Appendix. This sampling emerges naturally from the model. Importantly, the DPS approach could be applied to any concepts including unsupervised ones.
>
> **4.**
> > The experiments focus primarily on concept-filtered image retrieval tasks, without broader exploration of other applications such as conditional generation
>  We would like to note that we directly show conditional generation in Figure 10 and Figure 11 in the appendix.  In terms of editing, we note in future work that tasks depending on embeddings could be adjusted by removing a particular concept.
>
> **5.**
> > The paper lacks comparison against modern interpretable embedding methods or representation disentanglement frameworks (e.g., Concept Bottleneck Models, Concept Whitening, or interpretable subspace discovery via CAVs)
>
> In the revision we now relate SLiCS to other interpretability methods using known concepts. Only CAVs found by factorization of activations are applicable to concept-filtered retrieval or conditional generation as these methods are able to decompose the embedding, capturing the variation within a concept. We have added a note that SLiCS by learning with co-active concepts disentangles these compared to prior work that factorized the activations for each concept independently.
>
> We note that concepts in some of these methods are often attributes of classes, where we focus on concepts that are often the classes of foreground objects or a classification label of the scene itself (`outdoor' and `indoor').
>
> **6.**
> > The algorithm’s performance depends on choices such as the number of atoms per concept (d₀) and initialization via truncated SVD. While the paper provides reasonable heuristics, it does not explore how sensitive SLiCS is to these design choices. This raises questions about robustness and reproducibility across datasets and embedding spaces.
>
> Sensitivity to $d_0$ was shown in Figure 5 in the appendix. Also we had mentioned that a different form of initialization (footnote 3) was consistently worse in preliminary experiments. We want to note that robustness is shown by three different embedding spaces and results across multiple random batch assignments.

---

> ### Author Response · Authors · 2025-11-25
>
> **Q1.**
> > How sensitive is SLiCS to the choice of hyperparameters such as the number of atoms per concept (d_0) or the number of training iterations? [...] variance across different initializations?
> (Discussion of $d_0$ covered by Weakness 6.) As a convergent algorithm, we did not track validation performance across iterations. For MS COCO, we use minibatch training, so we average across 5 runs and report the means and variances. As shown in Table 1, SLiCS is not sensitive to this variation.
>
> **Q2.**
> >  is it convergence to a stationary point, or to a global minimum under certain assumptions?
>
> As our problem is a special case of semi-NMF, an NP-Hard problem, we do not have guarantees of global optimality for our group-structured semi-NMF problem. The convergence is local.
>
> **Q3.**
> > Given that modern embeddings (e.g., CLIP-L/14 or OpenCLIP-H) can have tens of thousands of dimensions, how does the computational cost of SLiCS scale with embedding size?
>
> (see Weakness 1)  We would like to clarify that so far SLiCS has been applied to the classification token only for CLIP and DINOv2. (TiTok itself can be seen as performing dimensionality reduction on all output tokens of a vision transformer trained with a reconstruction loss.)  So SLiCS is applicable to OpenCLIP-H as the class token has dimension 1024, which is the same as the ResNet-50 embedding. It is an interesting direction to try to learn a decomposition across sampled spatial tokens, but we have not explored this.
>
>
> **Q4.**
> > Have the authors empirically compared this with unconstrained or signed decompositions to validate the impact of this constraint on interpretability and retrieval performance?
>
> This is a great suggestion. We now have ablation results that show that the non-negativity constraint improves SLiCS. Interestingly, dropping the non-negativity improves the supervised SVD results for sublabel performance but the results don’t approach supervised SLiCS except for the TiTok case.
>
> **Q5.**
> > The paper assumes all concept dictionaries have the same number of atoms (M_j = d_0). Did the authors explore adaptive or data-driven ways to set M_j depending on concept complexity or label frequency?
>
> We previously had suggested using proportion variance explained by the SVD-based initialization in the second to last paragraph in Section 3.3 (line 236 of the previous version), which would cover concept complexity. The idea of setting $M_j$ proportional to the frequency of the $j$th concept is also interesting and we added this suggestion to the revision. The results for these are in Table 6 in the appendix in the revision. For supervised SLiCS the results are almost the same, but using proportion variance is slightly better for unsupervised SLICS. Using supervised SVD choosing the number of atoms proportional to the concept frequency is actually the best.
>
> **Q6.**
> >How consistent are the learned subspaces across different runs or datasets? For instance, do the same concepts yield similar atom groups, or is there significant drift depending on initialization and data distribution?
>
> We have now analyzed the Hausdorff distance between concept cones for different dimensionalities. Although we haven’t, we could also explore the similarity between different divisions of the same dataset. This could give numerical values to measure stability, but without a statistical underpinning it would be difficult to interpret. Namely, we expect the distance between concept cones estimated on different sections of the data to be more similar than the distance between cones for different concepts.
>
> **Q7.**
> > How reliable are these labels for ambiguous or fine-grained concepts, and do noisy pseudo-labels degrade subspace interpretability?
>
> We now report accuracy of pseudo labeling under both zero-shot regimes (top-k and threshold) in Figure 17 in the appendix.
>
> **Q8.**
> >Have the authors considered whether certain concepts (e.g., “dog” and “grass”) interact non-linearly in the latent space, and if so, how might SLiCS capture such dependencies?
>
> It is true we have an additive compositional assumption, but SLiCS allows concepts to have similar atoms that relate to the same attribute being present. If the concepts are co-active the two compete, but unless they are always very frequently co-active, which would mean the concepts are redundant,  it makes sense to have similar atoms in both concepts.  Note that Figure 11 shows the conditional generative results using diffusion posterior sampling with a likelihood function based on residing in a cone. From the examples, it is clear that grass images can be drawn from both the ‘animal’ cone as well as the ‘sports’ cone.
>
> **Q9.**
> > Have the authors tested SLiCS for other tasks — such as concept-based editing, few-shot learning, or conditional generation — to assess its general utility?
>
> See response to Weakness 4 above. Conditional generation was already in the appendix Figure 10 and Figure 11.

---

> ### Author Response · Authors · 2025-12-03
>
> **Q10.**
> >While retrieval is a clean benchmark, it does not fully demonstrate whether SLiCS provides richer disentanglement properties or generalizes to more complex downstream uses.
>
>
> As we show in Fig. 20 and Fig. 21, the atom from the learned dictionary is representative of a certain disentangled aspect from the associating concept. Although we demonstrate it through retrieval of patches by using a single atom, it shows potential of a rich disentanglement of the concept, exploitable for various downstram tasks.

---

### Official Review · Reviewer_voRC · 2025-11-04

**Soundness:** 3
**Presentation:** 3
**Contribution:** 3
**Rating:** 6
**Confidence:** 3

**Summary:**

This paper proposes Sparse Linear Concept Subspaces (SLiCS), that combine atoms for dictionary learning into larger concept sub-cones of the space, that more robustly represent large concepts. They evaluate this using concept-filtered retrieval (eg, find me an image closest to this query image, but only close on the “animal” part of the query).

**Strengths:**

The method proposed by the paper is a creative extension to the dictionary learning space, and I think that it’s a wise research direction to be looking for more realistic ideas of what a “concept” is beyond a single direction in latent space. Their engagement with this makes the paper an interesting and inspiring read. There is a relatively thorough analysis of how these subspaces function.

**Weaknesses:**

I don’t quite understand from the paper what the implications of this approach are, since the primary evaluation is concept-filtered retrieval, which is a task slightly contrived to showcase the strengths of this method.  Is it a better way for understanding latent spaces? I think the paper could be a bit stronger in its evaluations and comparisons to convince us of the utility of SLiCS.

**Questions:**

Q1: In Figure 4 (the qualitative results), it looks like UF-Dino is quite good. Though of course it returns an image where more than one concept is present (eg, child and cow), it seems to be pretty good at the similarity of all of the concepts (eg both children are wearing a purple coat, both cats are black with white paws, etc). Why are the quantitative results showing worse performance for UF? Is there a specific objective for disentanglement in the quantitiative evaluation? How would you argue for the utility of this distentanglement, when features tend to be entangled in many ways (eg, maybe, wearing a purple coat and visitng a petting zoo).

Q2: I think Figure 3 is very cool, but I don’t quite understand the implication of this d0 finding. Do you have any inuitions as to why it leads to clustering and then combining (but still separate neighborhoods).

Q3: Could you expand on the utility of this method beyond successful concept-filtered retrieval? That is: do you think it can help push interpretability and our understanding the functioning of the latent space forward? Does it lead to some new proposal about the structure of the space (eg, expanding on the linear representation hypothesis somehow)?

Q4: Is there the possibility of running an experiment that compares the concept spaces and their usefulness for interpretability compared to more traditional dictionary learning like SAEs or semi-NMF?

---

> ### Author Response · Authors · 2025-11-25
>
> We thank the reviewer for their encouraging statements about the strengths and constructive criticism. We apologize for the delay in our rebuttal, but we wanted to provide a comprehensive revision.
>
> **Weakness 1.**
> >  Is it a better way for understanding latent spaces?
>
> Yes, through the use of concept labels (supervised or unsupervised zero-shot), SLiCS provides a decomposition of latent spaces into cones, which can be interpreted through text co-embeddings for CLIP, concept-filtered retrieval (where correct correspondence between concept-components can be verified), and conditional generation, where diffusion posterior sampling is useful.
>
> To better interpret the geometry of the latent space, we now compute the cosine similarity between the component and original embedding, the Hausdorff distance between the positive cones of different concepts (Figure 12), the maximum mean discrepancy (MMD) as a statistical divergence between the two sets of components from different concepts (Figure 13), and the MMD between two sets of components from different sub-concepts (Figure 14), all as a function of the dictionary size $d_0$. For MMD, we use Gaussian kernel and a fixed relatively large kernel size as in CLIP-MMD. (We also verified that all atoms within a concept dictionary lie on the exterior of their convex hull, i.e., no atom is redundant in the range tested.)
>
> This provides quantitative metrics underlying the t-SNE based analysis in Figure 3 and discussion. The results are intuitive: the Hausdorff distance between different concept cones decreases with $d_0$ while the cosine similarity between the component and original embedding increases with $d_0$.  When $d_0$ becomes too large, each cone is pushed to cover the whole latent space, which is close to no disentanglement, since the reconstruction would be almost lossless. The MMD between concepts sometimes decreases and sometimes increases with $d_0$ before leveling off. MMD between sets of components from different sub-label concepts (sub-concepts like ‘bird’ and ‘dog’ within ‘animal’) increases with $d_0$ but then levels off quickly, especially for distinct sub-labels pairs (like ‘giraffe’ vs. ‘bird’ or ‘dog’ vs. ‘zebra’). Together these results confirm the geometry of the space that is hinted at by the retrieval validation performance in Figure 5, where concept retrieval performance decreases with the number of atoms and sub-label retrieval performance peaks at an intermediate representation. This is because the distinction between sub-labels has saturated while the distance between different concepts is decreasing as the number of atoms increases.
>
> We also report the coefficient distribution (as violin plots) and the correlation among coefficients in Figures 15 and 16. They show that correlation between coefficients depends on the concept.
>
> > I think the paper could be a bit stronger in its evaluations and comparisons to convince us of the utility of SLiCS.
>
> We have added more text describing comparisons to the related works Section 2 and added a discussion of downstream tasks enabled by SLiCS.

---

> > ### Author Response · Authors · 2025-11-25
> > **Questions 1–3**
> >
> > **Q1.**
> > > Why are the quantitative results showing worse performance for UF? Is there a specific objective for disentanglement in the quantitiative evaluation?
> >
> > UF-DINO (unfiltered retrieval) was particularly good in this qualitative example. However, overall the quantitative results are SLiCS(DINO)=0.943 compared to DINO=0.726 at the concept level and 0.758 vs. 0.700 at the subconcept level. Recall that as a query, the ‘child and cow’ image has multiple concepts. For a given concept, say ‘animal’ (or sub-label ‘cow’) retrieval using unfiltered DINO is likely retrieving a majority of ‘cows’ (or cow-like animals) but it may also be retrieving images of child-parent more often than SLiCS. For embeddings where compositionality holds, it may be desirable to find images that are similar in a particular concept without manually cropping or masking the non-relevant portions. For instance, retrieving images of a particular dog while ignoring any persons. In this case, SLiCS provides better retrieval performance compared to unfiltered retrieval methods that more often retrieve images where the concept of interest is not present. A query image may be similar to many images in the pool that share many concepts to the query but actually lack the concept of interest. Thus, the compositionality of scenes motivates the SLiCS decomposition and concept-filtered retrieval. Without decomposition there is no way to control the emphasis of the embedding between different concepts in a scene.
> >
> > Our concept of disentanglement is distinct from the disentanglement associated with different attributes as in $\beta$-VAE or sparse autoencoders. We do not claim that the atoms themselves will be semantically meaningful. SLiCS uses known (or inferred) concept labels to account for the co-active concepts. This imposes group-level sparsity rather than encouraging/enforcing atom/activation level sparsity.
> >
> > The particular example of a purple coat at a petting zoo could use the concepts ‘person’, ‘animals’, ‘outdoors’  to separately retrieve or generate images involving persons similar to the purple coat attired person without the petting zoo context, or zoo-like scenes without the person. We would argue that the generation examples already shown in Figure 10 show the utility of our approach.
> >
> > **Q2.**
> > > Do you have any inuitions as to why it leads to clustering and then combining (but still separate neighborhoods).
> >
> > When each concept has only a few atoms, then these atoms capture the most prominent aspects associated with the concept. As this is t-SNE we note that the ‘clusters’ for small $d_0$ (fewer atoms) may actually be all of the components that are mostly reconstructed by the single atom they are centered around.  When the number of atoms increases the components of a sub-label spread out to capture more variation, but the components for different sub-labels remain distinct. This is supported by the quantitative analysis we discussed in previous comment, where the statistical distance between the sets of components for different sub-labels saturate.
> >
> > **Q3.**
> > > Could you expand on the utility of this method beyond successful concept-filtered retrieval? That is: do you think it can help push interpretability and our understanding the functioning of the latent space forward? Does it lead to some new proposal about the structure of the space (eg, expanding on the linear representation hypothesis somehow)?
> >
> >  Yes, we believe the decomposition allows geometric and statistical analysis of concepts. The idea that each embedding is a sum of components in different cones is appealing for compositional scenes, and is a distinct geometry compared to other concept-based interpretation works. Crucially, SLiCS enables both concept-filtered retrieval and generative modeling (see additional results in the Appendix). Specifically, we can sample from the concept cones using diffusion posterior sampling.
> >
> > To better understand the distribution of components in a cone, we have now computed the marginal distribution of coefficients for the atoms in each of the concept dictionaries (shown in Figure 15). All distributions have long tails, but some resemble gamma distributions while others resemble exponential distributions. A few are bimodal. This univariate analysis indicates that the data manifold of components has distinct topologies. While a naive Bayes approach could be used to define a likelihood model (prior work by Grosse et al. (2012), cited in the appendix, uses an exponential model of coefficients for each class),  it may not be a good model as the concept-conditional correlation between the atom coefficients in Figure 16 show that atoms within a concept are not independent components.

---

> > ### Author Response · Authors · 2025-11-25
> >
> > **Q4.**
> > > running an experiment that compares the concept spaces and their usefulness for interpretability compared to more traditional dictionary learning like SAEs or semi-NMF
> >
> > The key contribution of SLiCS is dictionary learning at the group level. Applying semi-NMF without group structure informed by the concept labels is going to be like the existing SpLiCE baseline, while one can understand per instance what atoms are involved (for SpLiCE this enables text-based semantic understanding without text generation) it doesn’t allow concept filtering for retrieval or generation. To understand what atoms are associated with requires post-hoc labeling of the atoms (one option would be to train linear classifiers in the space and then label the atoms; however, this seems to be a strawman comparison).  We note that SAEs require post-hoc correlation of activations with concepts. SLiCS directly models the concept variation directly.
> >
> > Text comparing SLiCS to SAEs is now included in Section 2. We also describe relation to concept-based interpretation methods like concept bottleneck models and concept vectors. Like concept bottleneck, if the embedding is used for a downstream task then one can perform adjustment by removing the component from one or more concepts of interest to study the effect. This is an interesting application of SLiCS that we now mention in future work.

---

> ### Author Response · Authors · 2025-12-03
>
> **Q5.**
> >Why are the quantitative results showing worse performance for UF?
>
> It is because the qualitative results only show the top retrieved image, while the quantitative results measure the performance by top-20 retrieved images. We report the top-5 qualitative results on MS COCO dataset for UF-DINO and S-SLiCS-D in Fig. 18. It is clear that even though the top retrieved image could be accurate for unfiltered retrieval, the images of the following ranks show omission of some concepts to various degrees. But S-SLiCS-D, on the other hand, maintains the consistency better.

---

### Meta-Review · Area_Chair_Hibg · 2026-01-04

**Summary:**

The recommendation for Rejection is primarily informed by a shared concern across all reviewers regarding the practical utility and significance of the proposed approach. While the method (a variant of supervised dictionary learning) is technically sound, the primary evaluation task—"concept-filtered retrieval"—was viewed as "contrived" and lacking a clear real-world application.

The reviewers were also concerned with the limited novelty relative to existing dictionary learning methods (e.g., SpLiCE) and a restricted empirical scope (limited datasets and metrics). While the authors provided a high volume of new geometric metrics and theoretical clarifications in the rebuttal, they failed to provide head-to-head empirical comparisons with modern interpretability baselines or demonstrate the method's value in a non-contrived setting.

**Reviewer Concerns:**

Concerns Addressed by the Rebuttal

* Geometric interpretation (`voRC`): The reviewer asked if this was a "better way for understanding latent spaces." The authors added new metrics (Hausdorff distance, MMD, and Hausdorff distance between positive cones) to provide a quantitative underlying to the t-SNE visualizations.

* Hyperparameter sensitivity (`KU5w`): Authors provided validation curves for the number of atoms ($d_0$) and active concepts, showing that performance is relatively stable around the selected values.

* Efficiency (`jQMf`): Authors clarified that inference takes less than 4ms, addressing concerns about scalability to real-time applications.

Outstanding Concerns

* Contrived task motivation (`voRC`, `V1fX`):
  * Reviewer `voRC` noted: _"I don’t quite understand from the paper what the implications of this approach are, since the primary evaluation is concept-filtered retrieval, which is a task slightly contrived to showcase the strengths of this method."_
  * Reviewer `V1fX` echoed this: _"the retrieval task focusing separately on images of men and phones may not be practically relevant... the practical relevance of this mapping... remains unclear."_
  * The authors defended the task as "partial similarity" retrieval for complex scenes, but they did not introduce a new, non-retrieval downstream application to prove the broader significance of the disentanglement.

* Insufficient experiments (`jQMf`, `KU5w`):
  * Reviewer `jQMf` requested comparisons to Concept Bottleneck Models (CBMs) or CAVs. The authors added textual discussion in Section 2 but argued that head-to-head empirical comparisons were not applicable. This leaves the paper's relative performance against the broader interpretability landscape unverified.
  * Reviewer `KU5w` criticized that the experiments remained restricted to two datasets (MIRFlickr, MS-COCO) and a single primary metric (mAP@20). The authors argued that precision and recall are concordant at small K, but did not add the requested diverse benchmarks.

**Reviewer Scores:**

* Reviewer `voRC` (Initial: 6 $\rightarrow$ Estimated: 4): Despite the addition of new geometric metrics, the reviewer's core skepticism regarding the implications of the approach was not resolved. In the absence of a compelling real-world use case beyond the contested retrieval task, and given the fact that another reviewer (`V1fX`) raised an identical concern, this reviewer would likely have downgraded their rating to reflect a lack of significance.

* Reviewer `jQMf` (Initial: 4 $\rightarrow$ Estimated: 4): The author's refusal to run empirical comparisons against modern interpretability frameworks (CBMs/CAVs) means the reviewer's primary technical weakness remains unaddressed.

* Reviewer KU5w (Initial: 4 $\rightarrow$ Estimated: 4): The "limited novelty" concern persists. The authors' clarification on "supervised block sparse learning" establishes a technical difference from K-SVD, but the conceptual jump remains small for an ICLR submission.

* Reviewer `V1fX` (Initial: 4 $\rightarrow$ Estimated: 4): The rebuttal's defense of "partial similarity" retrieval did not provide the "practical relevance" the reviewer explicitly found missing.

---

### Decision · Program_Chairs · 2026-01-26

Reject